# Mitigating Reward Hacking in RLHF via Bayesian Non-negative Reward Modeling

Zhibin Duan [* 1]  Guowei Rong [* 2]  Zhuo Li [3 4]  Bo Chen [5]  Mingyuan Zhou [6]  Dandan Guo [# 2 7]

## Abstract

Reward models learned from human preferences are central to aligning large language models (LLMs) via reinforcement learning from human feedback, yet they are often vulnerable to reward hacking due to noisy annotations and systematic biases such as response length or style. We propose Bayesian Non-Negative Reward Model (BNRM), a principled reward modeling framework that integrates non-negative factor analysis into Bradley–Terry (BT) preference model. BNRM represents rewards through a sparse, non-negative latent factor generative process that operates at two complementary levels: instance-specific latent variables induce disentangled reward representations, while sparsity over global latent factors acts as an implicit debiasing mechanism that suppresses spurious correlations. Together, this disentanglement-then-debiasing structure enables robust uncertainty-aware reward learning. To scale BNRM to modern LLMs, we develop an amortized variational inference network conditioned on deep model representations, allowing efficient end-to-end training. Extensive empirical results demonstrate that BNRM substantially mitigates reward over-optimization, improves robustness under distribution shifts, and yields more interpretable reward decompositions than strong baselines.

---
[*]Equal contribution [#]Corresponding author. [1]School of Mathematics and Statistics, Xi'an Jiaotong University, Xi'an, Shaanxi, 710049, China. [2]School of Artificial Intelligence, Jilin University. [3]Shenzhen Research Institute of Big Data. [4]The Chinese University of Hong Kong, Shenzhen. [5]National Key Lab of Radar Signal Processing, Xidian University, Xi'an, Shaanxi 710071, China. [6]McCombs School of Business, The University of Texas at Austin, TX 78712, USA. [7]King Abdullah University of Science and Technology (KAUST) . Correspondence to: Dandan Guo <guodandan@jlu.edu.cn>.

*Proceedings of the 43rd International Conference on Machine Learning*, Seoul, South Korea. PMLR 306, 2026. Copyright 2026 by the author(s).

## 1. Introduction

With the advent of Large Language Models (LLMs), reinforcement learning from human feedback (RLHF) has emerged as a central paradigm for aligning model behavior with human values (Stiennon et al., 2020; Ouyang et al., 2022a; He et al., 2026; Wu et al., 2025). At the core of RLHF lies the reward models (RMs), which distill noisy human preference annotations into a differentiable signal for policy optimization (Ziegler et al., 2019; Ouyang et al., 2022a; Gao et al., 2023; Liu et al., 2024a). Despite the empirical success, learning reward models that are both reliable and generalizable remains a fundamental challenge (Casper et al., 2023; Touvron et al., 2023a; Liu et al., 2024a; Li et al., 2025b). A prevalent failure mode, commonly referred to as reward over-optimization or reward hacking, occurs when the policy exploits spurious correlations encoded in the proxy RM, yielding behavior that scores highly under the learned reward but deviates from true human objectives (Gao et al., 2023; Coste et al., 2024).

A primary source of reward over-optimization in reward modeling is reward misgeneralization (Casper et al., 2023; Miao et al., 2024), whereby RMs extrapolate incorrectly beyond the training distribution and thus form poor proxies for true human preferences, which arises from two interacting factors: on the one hand, noisy, subjective, and heterogeneous human annotations (Gao et al., 2024); on the other hand, the tendency of deep neural networks to exploit shortcut features and learn spurious correlations that bypass their intended semantic objectives (Geirhos et al., 2020; Zhang et al., 2016). As a result, RMs often overemphasize superficial cues, like response length (Singhal et al., 2023), phrasing patterns, or stylistic artifacts (Zhang et al., 2025; Miao et al., 2024; Wang et al., 2024), which are easy to optimize but misaligned with genuine human intent, ultimately undermining the reliability and safety of RLHF (Casper et al., 2023; Gao et al., 2023).

To mitigate reward hacking, recent work has explored Bayesian principles to alleviate overfitting in highly over-parameterized reward models (Wang & Yeung, 2016; Wilson & Izmailov, 2020). Practical approaches such as reward model ensembling (Lakshminarayanan et al., 2017;

Coste et al., 2024; Eisenstein et al., 2023; Ramé et al., 2024; Zhang et al., 2024a) improve robustness by aggregating multiple predictors, but at the cost of training and maintaining several large-scale models, resulting in substantial computational and memory overhead. Beyond ensembles, information-theoretic methods introduce variational information bottleneck objectives to suppress spurious latent features (Miao et al., 2024; Li et al., 2025a). However, these approaches rely on implicit notions of relevance and often fail to explicitly disentangle semantic intent from nuisance factors (Yang et al., 2022). Other lines of work address robustness by correcting specific biases through supervised interventions (Chen et al., 2024), most notably response-length bias, but such methods typically generalize poorly beyond narrowly defined settings.

To address these limitations more fundamentally, we revisit sparsity-aware Bayesian models (SBMs) (Wipf & Rao, 2004), such as non-negative factor analysis (NFA) (Blei et al., 2003; Zhou et al., 2012). These models offer two key advantages. First, probabilistic latent factor modeling enables explicit treatment of both epistemic and aleatoric uncertainty, which is essential when learning from noisy and inconsistent human feedback (Gao et al., 2023). Second, sparsity acts as an implicit regularizer that discourages reliance on spurious or non-invariant features, thereby improving robustness to shortcut correlations (Zhou et al., 2022). Moreover, the inherent non-negativity constraints in these models induce disentangled, parts-based representations, yielding substantially improved interpretability compared to dense and opaque embeddings (Lee & Seung, 1999; Nguyen et al., 2016).

Inspired by these insights, we revisit RMs from an NFA perspective and propose the Bayesian Non-Negative Reward Model (BNRM). BNRM integrates the probabilistic structure of NFA with the expressive representations of large language models, formulating reward learning as a stochastic generative process over latent, non-negative reward factors that explicitly capture uncertainty. Besides, as illustrated in Figure 1, BNRM departs from conventional dense reward functions by imposing sparsity-aware structure: local sparsity promotes disentangled representations of semantic preference factors, while global sparsity suppresses spurious correlations and facilitates systematic debiasing. To scale BNRM to modern LLMs, we develop an amortized variational inference network conditioned on backbone representations. By parameterizing the variational posterior with a reparameterizable Weibull distribution (Zhang et al., 2020), BNRM enables end-to-end training via standard backpropagation. Extensive experiments demonstrate that BNRM substantially improves robustness to reward over-optimization, enhances interpretability, and generalizes better under distribution shifts. Our contributions are summarized as follows:

- We propose **BNRM**, a Bayesian non-negative reward modeling framework that jointly enforces sparsity and models uncertainty to mitigate reward hacking.

- We introduce a scalable amortized variational inference scheme with reparameterizable Weibull posteriors, enabling efficient integration with large language models.

- We empirically show that BNRM outperforms strong baselines in robustness, interpretability, and out-of-distribution generalization.

## 2. Related Work

**Mitigating Reward Overoptimization in RLHF.** Reward models are prone to overfitting on training data, leading to reward overoptimization where policies exploit the learned proxy instead of the true human objective (Gao et al., 2023). Existing mitigation efforts can be broadly categorized. One major line of work focuses on *uncertainty-aware modeling*, primarily through computationally intensive model ensembles or by adding post-hoc uncertainty penalties to the reward signal (Coste et al., 2024; Eisenstein et al., 2023; Lin et al., 2023; Zhang et al., 2024b; Yang et al., 2024a; Lou et al., 2024; Sun et al., 2025; Li et al., 2026). Another category involves *data and policy-level regularization*, using techniques like label smoothing, adaptive margin, and adding SFT-based constraints during policy optimization (Wang et al., 2024; Liu et al., 2024c; Li et al., 2025b). While these methods effectively mitigate the symptoms, our work posits that the problem's root is the reward model's reliance on dense, non-interpretable features. We therefore propose a fundamentally different approach that reshapes the reward representation itself by injecting the principles of Non-Negative Factor Analysis (NFA) into the reward modeling process, allowing us to directly learn a sparse and robust reward function that is inherently less susceptible to spurious correlations.

**Non-negative Factor Analysis.** NFA models, such as latent Dirichlet allocation (Blei et al., 2003) and Poisson factor analysis (Zhou et al., 2012), are classical probabilistic tools for learning parts-based representations from data. Their ability to uncover sparse latent structures has made them highly effective for tasks like topic modeling (Zhou et al., 2015). While recent work has explored integrating NFA's benefits into general deep neural networks (Duan et al., 2021; 2026; Hu et al., 2025), our contribution is the principled synthesis of NFA's generative structure within the modern RLHF pipeline. Distinct from prior studies, we leverage NFA's sparse inductive bias specifically to combat reward over-optimization (Gao et al., 2023; Zhou et al., 2022) and enhance robustness against distributional shifts (Wilson & Izmailov, 2020).

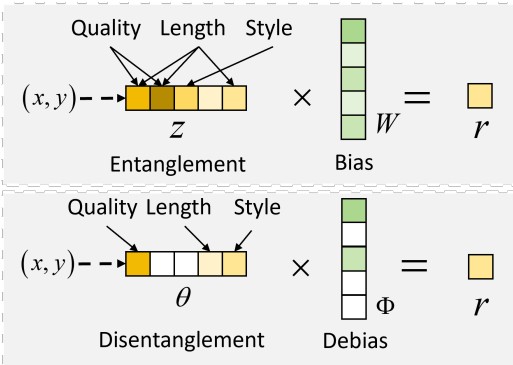

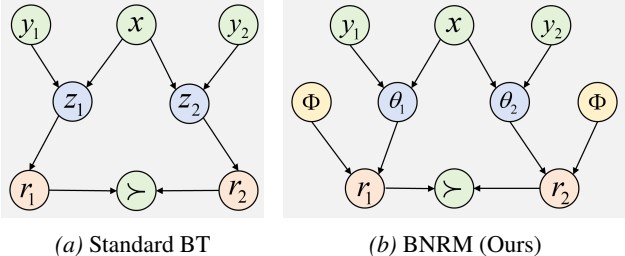

*(a)* Standard BT        *(b)* BNRM (Ours)

*Figure 2.* Graphical model representations. (a) The standard BT model; (b) our proposed BNRM. Here, $\mathbf{x}$ denotes the prompt, $\mathbf{y}_1, \mathbf{y}_2$ are candidate responses, and nodes represent the predictive process for the preference $\mathbf{y}_1 \succ \mathbf{y}_2$.

*Figure 1.* Motivation on disentanglement and debiasing for alleviating spurious correlations in reward modeling.

## 3. Preliminaries

### 3.1. Scalar Preference Reward Model in RLHF

A reward model (RM) aims at learning human preferences based a parameterized model $r_\phi$ on a human preference dataset $\mathcal{D} = \{(\boldsymbol{x}, \boldsymbol{y}_1, \boldsymbol{y}_2)\}$, where $\boldsymbol{x}$ is a user input prompt, $\boldsymbol{y}_1$ and $\boldsymbol{y}_2$ are the preferred (chosen) and non-preferred (rejected) responses, by minimizing a ranking loss following the Bradley-Terry (BT) objective (Bradley & Terry, 1952):

$$p(\boldsymbol{y}_1 \succ \boldsymbol{y}_2 \,|\, \boldsymbol{x}) = \sigma(r_\phi(\boldsymbol{x}, \boldsymbol{y}_1) - r_\phi(\boldsymbol{x}, \boldsymbol{y}_2)),$$

where $\sigma(\cdot)$ is the Sigmoid function. The optimized reward model serves as a proxy for human preferences, enabling the subsequent RL fine-tuning phase. $r_\phi$ typically consists of a backbone feature extractor $f$ and a final linear head $W_{\text{bt}}$ (Stiennon et al., 2022; Ouyang et al., 2022b; Yang et al., 2024b), where the backbone $f$ is often initialized with an pretrained LLM parameter $W_{\text{llm}}$ and projects the prompt and the corresponding response into a factor vector, denoted as $\boldsymbol{z} = f(\boldsymbol{x}, \boldsymbol{y}) \in \mathbb{R}^{1 \times d_{\text{model}}}$, where $d_{\text{model}}$ is hidden dimension. A new linear head, $W_{\text{bt}} \in \mathbb{R}^{d_{\text{model}} \times 1}$, is then added to project the feature representation $\boldsymbol{z}$ into a scalar reward value, expressed as

$$r_\phi(x, y) = \boldsymbol{z}^T W_{\text{bt}}.$$

The training of the RM involves fine-tuning the parameters of $f$ and $W_{\text{bt}}$ on the preference dataset $\mathcal{D}$. However, this standard implementation suffers from two fundamental limitations: **Deterministic and Overconfident Scoring**: The model produces a single, deterministic reward, failing to capture uncertainty in human preferences and leading to over-optimization. **Specious Correction**: The dense, "black-box" nature of the features $\boldsymbol{z}$ and weights $W$ makes the model prone to reward hacking by exploiting spurious correlations, as shown in Fig.1(Top). The ultimate goal is to use this learned reward signal $r_\phi$ to optimize the policy in the final RL stage. A flawed, deterministic, and easily hackable RM inevitably leads to a misaligned policy, un-

derscoring the necessity for a more robust reward modeling framework, which we introduce in the subsequent sections.

### 3.2. Non-negative Factor Analysis (NFA)

Non-negative factor analysis methods, such as Poisson factor analysis (PFA) (Zhou et al., 2012), are widely used as topic models (Blei & Lafferty, 2009). They impose non-negative stochastic latent variables on the model parameters to learn interpretable, parts-based representations of data. Specifically, representing document $\boldsymbol{x}$ as a BOW vector $\boldsymbol{b} \in \mathbb{Z}^V$, where $\mathbb{Z} = \{0, 1, \cdots\}$ and $V$ is the vocabulary size, PFA models $\boldsymbol{b}$ under the Poisson likelihood as

$$\boldsymbol{b} \sim \text{Poisson}\left(\boldsymbol{\Phi}\boldsymbol{\theta}\right), \boldsymbol{\theta} \sim \text{Gamma}\left(\alpha, 1\right). \quad (1)$$

Here, the matrix $\boldsymbol{\Phi} \in \mathbb{R}_+^{V \times K}$ is the dictionary, where each column represents a topic as a distribution over words. The vector $\boldsymbol{\theta} \in \mathbb{R}_+^K$ contains the document-specific topic proportions (*i.e.*, document features) that represent the strength of each topic in the document. Benefiting from the sparsity of latent variables, which is often encouraged by placing Gamma priors with hyperparameter $\alpha$ on $\boldsymbol{\theta}$, NFA effectively handles overdispersed data and exhibits strong generalization ability.

## 4. The Bayesian Non-negative Reward Model

Our methodology is rooted in a Bayesian perspective, comprising a generative process that describes our idealized assumptions about how preferences are formed, and an inference process that details how we approximate this model in practice using deep neural networks.

### 4.1. BT from a Bayesian Viewpoint

The BNRM framework, introduced intuitively in the previous section, can be formally derived as a hierarchical Bayesian extension of the standard Bradley-Terry (BT) model (Bradley & Terry, 1952). This viewpoint clarifies how BNRM systematically addresses the limitations of the deterministic approach. From a Bayesian perspective, as

shown in Figure 2a, the standard BT model is a special case of the following integral formulation:

$$p(\boldsymbol{y}_1 \succ \boldsymbol{y}_2 \,|\, \boldsymbol{x}, \boldsymbol{y}_1, \boldsymbol{y}_2) = \int_{\boldsymbol{z}_1, \boldsymbol{z}_2} p(\boldsymbol{y}_1 \succ \boldsymbol{y}_2 \,|\, \boldsymbol{z}_1, \boldsymbol{z}_2)$$
$$p(\boldsymbol{z}_1 \,|\, \boldsymbol{y}_1, \boldsymbol{x}) \, p(\boldsymbol{z}_2 \,|\, \boldsymbol{y}_2, \boldsymbol{x}) \, d\boldsymbol{z}_1 \, d\boldsymbol{z}_2. \quad (2)$$

Here, the deterministic nature of the underlying neural network $f$ means that the conditional probability $P(\boldsymbol{z} \,|\, \boldsymbol{y}, \boldsymbol{x})$ is a Dirac delta distribution, $P(\boldsymbol{z} \,|\, \boldsymbol{y}, \boldsymbol{x}) = \delta(\boldsymbol{z} - f(\boldsymbol{x}, \boldsymbol{y}))$, which offers no mechanism to capture uncertainty. Our BNRM generalizes this model in two steps:

**Modeling Aleatoric Uncertainty.** We first replace the deterministic latent representation $z$ with a stochastic latent variable $\theta$. This allows the model to capture the inherent randomness and ambiguity in human preference data. The preference probability is now marginalized over the distribution of $\theta$:

$$p(\boldsymbol{y}_1 \succ \boldsymbol{y}_2 \,|\, \boldsymbol{x}, \boldsymbol{y}_1, \boldsymbol{y}_2) = \int_{\theta_1, \theta_2} P(\boldsymbol{y}_1 \succ \boldsymbol{y}_2 \,|\, \theta_1, \theta_2)$$
$$p(\theta_1 \,|\, \boldsymbol{y}_1, \boldsymbol{x}) \cdot P(\theta_2 \,|\, \boldsymbol{y}_2, \boldsymbol{x}) \, d\theta_1 \, d\theta_2. \quad (3)$$

**Modeling Epistemic Uncertainty.** To further account for the model's own uncertainty about the global reward factors, we treat the final layer weights (denoted as $\Phi$) as a global stochastic variable. This leads to the full, formal generative process for BNRM, as shown in Figure 2b:

$$p(\boldsymbol{y}_1 \succ \boldsymbol{y}_2 \,|\, \boldsymbol{x}, \boldsymbol{y}_1, \boldsymbol{y}_2) = \int_{\theta_1, \theta_2, \Phi} p(\boldsymbol{y}_1 \succ \boldsymbol{y}_2 \,|\, \theta_1, \theta_2, \Phi)$$
$$p(\theta_1 \,|\, \boldsymbol{y}_1, \boldsymbol{x}) \, p(\theta_2 \,|\, \boldsymbol{y}_2, \boldsymbol{x}) \, p(\Phi) \, d\theta_1 \, d\theta_2 \, d\Phi, \quad (4)$$

which final integral represents the complete generative process of our proposed model.

## 4.2. The BNRM Generative Process

Recall that Eq. 4 is designed to systematically capture both aleatoric and epistemic uncertainty in reward modeling. Building on this formulation, we propose a fully probabilistic generative model in which human preferences arise from sparse, non-negative latent factors. This framework replaces the standard deterministic reward formulation, $r = f(\boldsymbol{x}, \boldsymbol{y})^\top W_{\mathrm{bt}}$, with a structured Bayesian alternative that explicitly models uncertainty, disentanglement, and bias. The generative process introduces two complementary sets of latent variables:

**1. Local Sparse Representation (Disentanglement).** For each prompt–response pair $(\boldsymbol{x}, \boldsymbol{y})$, we introduce a non-negative latent vector $\boldsymbol{\theta} \in \mathbb{R}_+^K$ that captures the sparse activations of global reward factors specific to $\boldsymbol{y}$. Sparsity in $\boldsymbol{\theta}$ encourages *instance-level disentanglement* by activating only a small subset of latent factors, thereby improving

identifiability and reducing reliance on entangled or spurious features (Zheng et al., 2022).

**2. Global Reward Dictionary (Debiasing).** We further introduce a global dictionary of reward factors $\Phi \in \mathbb{R}_+^K$, shared across all data points, which defines a universal non-negative basis for evaluating response quality. Sparsity imposed on $\Phi$ acts as a *population-level regularizer*, selectively retaining invariant and semantically meaningful factors while suppressing spurious correlations in reward estimation (Zhang et al., 2021; Zhou et al., 2022).

To jointly enforce non-negativity and sparsity, we place Gamma priors, a standard choice in non-negative factor analysis (Zhou et al., 2012), on both sets of latent variables:

$$\Phi \sim \mathrm{Gamma}(\gamma_0, \delta_0), \qquad \boldsymbol{\theta} \sim \mathrm{Gamma}(\alpha_0, \beta_0). \quad (5)$$

Given these latent factors, the scalar reward associated with a response $(\boldsymbol{x}, \boldsymbol{y})$ is generated as

$$r(\boldsymbol{x}, \boldsymbol{y}) = \boldsymbol{\theta}^\top \Phi, \quad (6)$$

which yields a non-negative, interpretable reward constructed from sparse factor activations. Finally, for a pair of candidate responses $(\boldsymbol{y}_1, \boldsymbol{y}_2)$ with corresponding rewards $(r_1, r_2)$, the observed human preference is generated via a Bradley–Terry likelihood:

$$p(\boldsymbol{y}_1 \succ \boldsymbol{y}_2 \,|\, r_1, r_2) = \sigma(r_1 - r_2), \quad (7)$$

where $\sigma(\cdot)$ denotes the logistic sigmoid function. This completes a coherent probabilistic account of how human preferences emerge from sparse, non-negative latent reward factors, naturally supporting uncertainty quantification, disentanglement, and debiasing within a unified Bayesian framework.

## 4.3. Variational Inference and Training Objective

Given the generative formulation of BNRM defined in Eqs. 4 and 5, the exact posterior distributions over the latent variables, $p(\boldsymbol{\theta} \,|\, \boldsymbol{x}, \boldsymbol{y})$ and $p(\Phi \,|\, \mathcal{D})$, are analytically intractable. We therefore resort to variational inference (VI) to obtain a scalable approximation. Specifically, we introduce tractable variational distributions $q(\boldsymbol{\theta} \,|\, \boldsymbol{x}, \boldsymbol{y})$ and $q(\Phi)$ to approximate the true posteriors. Importantly, we re-purpose the deep LLM backbone $f$ not as part of the generative model, but as a powerful *inference network* (encoder) for amortized inference of the local latent variables. As illustrated in Figure 3, for each prompt–response pair $(\boldsymbol{x}, \boldsymbol{y})$, the inference proceeds as follows:

We first extract a deterministic, high-dimensional and dense feature representation using the LLM backbone with parameter $W_{\mathrm{llm}}$: $\boldsymbol{z} = f(\boldsymbol{x}, \boldsymbol{y}) \in \mathbb{R}^{d_{\mathrm{model}}}$. The feature vector $\boldsymbol{z}$ is then used to parameterize the variational distribution of

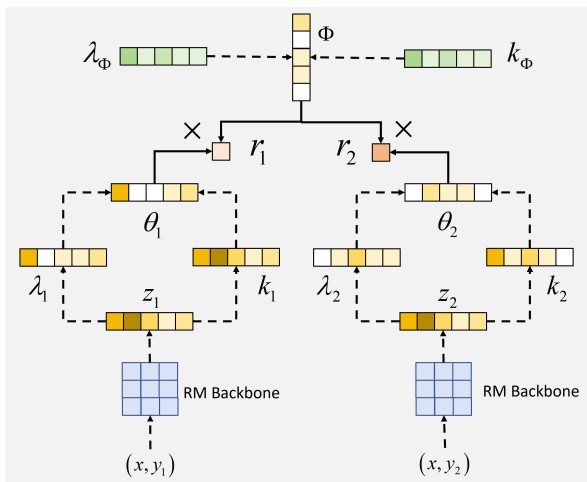

*Figure 3.* Variational Inferencer for BNRM.

the corresponding local latent variable $\boldsymbol{\theta}_i$. Following prior work on scalable inference for non-negative latent variable models (Zhang et al., 2020), we adopt a Weibull distribution due to its convenient reparameterization and its ability to model sparse, positive random variables:

$$q(\boldsymbol{\theta} \mid \boldsymbol{x}, \boldsymbol{y}) = \text{Weibull}(\boldsymbol{k}, \boldsymbol{\lambda}),$$
$$(\boldsymbol{k}, \boldsymbol{\lambda}) = \text{Activation}(\boldsymbol{z}W_{\text{vi}}), \qquad (8)$$

where $W_{\text{vi}} \in \mathbb{R}^{d_{\text{model}} \times 2K}$, $\boldsymbol{k}, \boldsymbol{\lambda} \in \mathbb{R}_+^K$ denote the shape and scale parameters, respectively. Within the inference network, we parameterize the shape parameter $\boldsymbol{k}_i$ using a `Softplus` activation to ensure differentiability and numerical stability, while the scale parameter $\boldsymbol{\lambda}_i$ is parameterized using a `ReLU` activation, which empirically encourages sparsity in the sampled latent variables.

The variational distribution for the global latent variable $q(\Phi)$ is parameterized analogously as a Weibull distribution with its own set of learnable parameters $W_\Phi$. We train the entire model, including the LLM backbone $f$ and the parameters of all variational distributions, by maximizing the evidence lower bound (ELBO) on the log-likelihood of the preference dataset $\mathcal{D}$:

$$\max_{W_{\text{llm}}, W_{\text{vi}}, \Phi} \mathcal{L}(\mathcal{D}) = \max_{W_{\text{llm}}, W_{\text{vi}}, \Phi} \Big[ \mathbb{E}_{q(\boldsymbol{\theta} \mid \boldsymbol{x}, \boldsymbol{y})q(\Phi)} \big[ \log p(\mathcal{D} \mid \boldsymbol{\theta}, \Phi) \big]$$
$$- \eta \text{KL}(q(\boldsymbol{\theta} \mid \boldsymbol{x}, \boldsymbol{y}) \, \| \, p(\boldsymbol{\theta}))$$
$$- \eta \text{KL}(q(\Phi) \, \| \, p(\Phi)) \Big]. \qquad (9)$$

The first term corresponds to the reconstruction likelihood, ensuring that the inferred latent variables explain the observed human preferences. The KL divergence regularizes the variational posteriors toward the respective priors, thereby enforcing sparsity, controlling model complexity, and improving robustness against reward overoptimization. $\eta$ is a trade-off balancing likelihood and KL divergence. We summarize the algorithm in Appendix A.1.

## 4.4. Intuition on Disentanglement and then Debias for Reward Modeling

We provide an intuitive analysis of why BNRA effectively mitigates spurious correlations in reward modeling. As illustrated in Figure 1 (bottom), sparsity constraints imposed on the local and global latent variables play complementary and synergistic roles. Specifically, sparsity in the local latent variables $\boldsymbol{\theta}$ encourages *instance-level disentanglement* by activating only a small subset of semantic factors that are sufficient to explain a given prompt–response pair. This selective activation suppresses incidental or idiosyncratic features, thereby improving interpretability and reducing sensitivity to spurious patterns at the individual sample level. In contrast, sparsity in the global latent variables $\Phi$ enforces *population-level invariance*, effectively identifying and down-weighting systematic but non-causal biases that persist across the dataset. Crucially, the interaction between local and global sparsity yields a structured reward function that is both interpretable and robust.

## 5. Experiment

### 5.1. Experiment Setup

**Training and Evaluation Datasets.** Following Yang et al. (2024b), our BNRM was trained on the Unified-feedback (UF) preference dataset that contains diverse human preference annotations that cover a wide range of dialogue scenarios and reward signals. Specifically, we randomly sampled 40K and 400K samples from the UF dataset for model training, and assessed the performance on a held-out 8K validation split. In addition, we further evaluated the models on several out-of-distribution (OOD) datasets, including RM-Bench (Liu et al., 2024b), RewardBench (Lambert et al., 2024), HHH-Alignment (Askell et al., 2021), and MT-Bench (Zheng et al., 2023), which better simulate real-world scenarios. Besides, we also consider Skywork-Preference-v0.2 (SP) training dataset (Liu et al., 2025) and compare it with advanced reward modeling approaches. The code is available at `https://github.com/GuoweiRong/Bayesian-Non-negative-Reward-Model`.

**Base Models and Training Details.** In reward modeling experiments, we firstly used gemma-2b-it (Gemma, 2024b) and gemma-2-2b-it (Gemma, 2024a) as base models, accelerated by Low-Rank Adaptation (LoRA) (Hu et al., 2021) for 2 epochs. In addition, we fully fine-tune the larger Skywork-Reward-Llama-3.1-8B (Liu et al., 2024a) reward model on the Skywork-Preference-v0.2 (SP) dataset within 1 epoch. Best-of-N (BoN) sampling test was performed exclusively with the two Gemma models, and our BNRM obtained via LoRA as proxy reward models. In the Proximal Policy Optimization (PPO) (Schulman et al., 2017) of real

*Table 1.* Results on ID and OOD evaluation with 40/400K UF training examples using LoRA. Best is **bold** and second-best is underlined. BT-BNRM and GRM-BNRM are based on BT and GRM-SFT, respectively. UF, HHH, and MT denote Unified Feedback, HHH Alignment, MT Bench, respectively.

| Reward Model | Gemma 2B it | | | | | | | | Gemma2 2B it | | | | | | | |
|---|---|---|---|---|---|---|---|---|---|---|---|---|---|---|---|---|
| | UF | HHH | MT | RewardBench | | | | | UF | HHH | MT | RewardBench | | | | |
| | | | | Average | Chat | Chat-Hard | Safety | Reasoning | | | | Average | Chat | Chat-Hard | Safety | Reasoning |
| *Unified-Feedback 40K* | | | | | | | | | | | | | | | | |
| BT | 68.8 | 70.3 | 69.1 | 64.5 | 95.8 | 37.3 | 59.9 | 64.8 | 74.5 | 84.2 | 73.3 | 75.7 | 96.1 | 50.7 | 80.9 | 75.0 |
| BT-Margin | 69.6 | 69.8 | 71.0 | 66.1 | 97.2 | 37.5 | 56.8 | 72.7 | 74.7 | 83.6 | 75.1 | 72.9 | 97.0 | 49.7 | 80.4 | 64.6 |
| BT-LabelSmooth | 68.5 | 68.8 | 71.9 | 61.1 | 91.6 | 39.0 | 53.8 | 60.2 | 74.7 | 81.5 | 74.7 | 76.6 | 96.8 | 51.8 | 82.3 | 75.3 |
| BT-Ensemble | 69.9 | 72.2 | 71.1 | 65.2 | 96.1 | 38.2 | 58.8 | 67.6 | 75.1 | 84.9 | 74.3 | 77.8 | 98.0 | 49.3 | 81.1 | 82.8 |
| GRM-DPO | 70.2 | 71.6 | 71.3 | 70.8 | 97.8 | 42.1 | 77.9 | 65.2 | 75.5 | 85.3 | 74.4 | 77.6 | 98.0 | 50.4 | 82.2 | 79.8 |
| GRM-DPO-Noref | 71.4 | 76.6 | 72.1 | 66.6 | 92.5 | 39.9 | 72.5 | 61.4 | 75.2 | 83.5 | 74.5 | 77.5 | 98.0 | 51.1 | 81.9 | 79.0 |
| GRM-SFT | 71.5 | 78.7 | 73.0 | 66.8 | 94.1 | 41.9 | 69.5 | 61.5 | 75.8 | 85.5 | 74.2 | 77.3 | 96.4 | 50.0 | 84.3 | 78.5 |
| InfoRM | 71.6 | 83.9 | 71.4 | 71.2 | 95.8 | 43.2 | 79.5 | 66.3 | 73.9 | 83.9 | 74.6 | 79.2 | 97.8 | 57.0 | 85.8 | 76.1 |
| BT-BNRM | 74.2↑5.4 | 83.6↑13.3 | 75.2↑6.1 | 72.5↑8.0 | 95.6↓0.2 | 43.3↑6.0 | 80.9↑21.0 | 70.1↑5.3 | 77.2↑2.7 | 87.8↑3.6 | 76.8↑3.5 | 79.7↑4.0 | 97.1↑1.0 | 56.3↑5.6 | 85.3↑4.4 | 79.9↑4.9 |
| GRM-BNRM | 74.1↑2.6 | 82.4↑3.7 | 75.1↑2.1 | 71.8↑5.0 | 95.7↑1.6 | 41.6↓0.3 | 81.5↑12.0 | 68.4↑6.9 | 76.9↑1.1 | 85.1↓0.4 | 76.0↑1.8 | 80.5↑3.2 | 97.5↑1.1 | 54.3↑4.3 | 86.2↑1.9 | 84.1↑5.6 |
| *Unified-Feedback 400K* | | | | | | | | | | | | | | | | |
| BT | 72.1 | 73.4 | 71.2 | 68.2 | 95.5 | 38.0 | 73.8 | 65.3 | 76.6 | 86.4 | 75.2 | 77.5 | 97.2 | 51.4 | 83.2 | 78.3 |
| BT-Margin | 72.0 | 75.0 | 72.6 | 70.2 | 95.8 | 38.4 | 73.9 | 72.5 | 77.3 | 85.9 | 76.0 | 74.6 | 97.5 | 48.9 | 83.8 | 68.3 |
| BT-LabelSmooth | 71.5 | 72.1 | 71.2 | 70.6 | 94.4 | 37.3 | 73.2 | 77.4 | 76.6 | 85.4 | 75.4 | 79.2 | 98.0 | 52.4 | 82.4 | 83.9 |
| BT-Ensemble | 72.8 | 76.8 | 73.7 | 67.0 | 96.4 | 38.4 | 73.8 | 59.5 | 76.9 | 83.9 | 76.3 | 78.2 | 97.8 | 48.5 | 83.8 | 82.9 |
| GRM-DPO | 73.8 | 79.2 | 73.4 | 68.2 | 95.3 | 39.0 | 77.8 | 60.6 | 77.3 | 87.1 | 76.2 | 78.9 | 98.9 | 48.2 | 83.4 | 85.2 |
| GRM-DPO-Noref | 73.9 | 79.7 | 73.0 | 70.2 | 95.8 | 40.1 | 78.7 | 66.2 | 76.7 | 87.5 | 75.3 | 79.3 | 98.0 | 49.6 | 85.4 | 84.0 |
| GRM-SFT | 73.2 | 79.8 | 73.4 | 70.8 | 97.8 | 42.1 | 77.9 | 65.2 | 78.9 | 88.2 | 77.5 | 77.7 | 97.9 | 50.8 | 84.6 | 77.6 |
| InfoRM | 76.2 | 85.4 | 74.6 | 72.7 | 97.2 | 44.5 | 78.1 | 70.8 | 77.3 | 85.4 | 76.3 | 80.7 | 98.0 | 57.0 | 85.9 | 81.8 |
| BT-BNRM | 77.0↑4.9 | 86.4↑13.0 | 76.1↑4.9 | 73.2↑5.0 | 96.4↑0.9 | 41.7↑3.7 | 81.8↑8.0 | 72.9↑7.6 | 78.8↑2.2 | 88.2↑1.8 | 78.2↑3.0 | 79.5↑2.0 | 97.5↑0.3 | 51.7↑0.3 | 84.9↑1.7 | 83.8↑5.5 |
| GRM-BNRM | 76.6↑3.4 | 84.6↑4.8 | 76.9↑3.5 | 71.3↑0.5 | 95.7↓2.1 | 42.1↑0.0 | 80.8↑2.9 | 66.7↑1.5 | 78.7↓0.2 | 88.2↑0.0 | 78.0↑0.5 | 79.4↑1.7 | 97.4↓0.5 | 52.9↑2.1 | 85.1↑0.5 | 82.3↑4.7 |

*Table 2.* RewardBench performance comparison over baselines, including both generative and discriminative reward models. The best performance is highlighted in **bold**, and we cite baseline results from Liu et al. (2024b).

| | Method | Average | Chat | Chat-Hard | Safety | Reasoning |
|---|---|---|---|---|---|---|
| Generative | SFR-LLaMa-3.1-70B-Judge-I | 92.7 | 96.9 | 84.8 | 91.6 | 97.6 |
| | Gemini-1.5 | 86.8 | 94.1 | 77.0 | 85.8 | 90.2 |
| | GPT-4o | 86.7 | 96.1 | 76.1 | 88.1 | 86.6 |
| | SFR-nemo-12B-Judge-r | 90.3 | 97.2 | 82.2 | 86.5 | 95.1 |
| Discriminative | Nemotron-340B-Reward | 92.2 | 95.8 | 87.1 | 92.2 | 93.6 |
| | ArmoRM-Llama3-8B-v0.1 | 90.8 | 96.9 | 76.8 | 92.2 | 97.3 |
| | InternLM-20B-Reward | 90.2 | 98.9 | 76.5 | 89.9 | 95.8 |
| | Llama-3-OffsetBias-RM-8B | 89.4 | 97.2 | 81.8 | 86.8 | 91.9 |
| | Skywork-1-1BT-RM-8B | 91.8 | 94.6 | 84.5 | 91.5 | 96.5 |
| | Skywork-Reward-Llama-3.1-8B | 93.1 | 94.7 | 88.4 | 92.7 | 96.5 |
| | BNBT-Reward-Llama-3.1-8B | **93.6**↑0.5 | 95.3↑0.6 | **89.7**↑1.3 | 92.6↓0.1 | 96.9↑0.2 |

LLM fine-tuning, we applied LoRA to fine-tune Llama3.1-8B-Instruct (Meta, 2024) and OpenRLHF-Llama3-8B-SFT (Dong et al., 2024) for 1 epoch. More detailed training configurations are provided in Appendix C.

**Baselines.** As detailed in Appendix C.4, we consider the following baselines: (1) BT and its variants, including BT (Bradley & Terry, 1952), BT-Margin (Touvron et al., 2023b), BT-Frozen, BT-Ensemble (Coste et al., 2024), and BT-Label Smoothing (Wang et al., 2024); (2) GRM (Yang et al., 2024b) ; (3) InfoRM (Miao et al., 2024).

### 5.2. Evaluation on Reward Modeling

**ID and OOD Evaluation.** Table 1 clearly shows that BNRM consistently boosts the corresponding base reward models and further outperforms advanced baselines across most ID and OOD evaluation tasks, regardless of training data scales. For example, with 40K training exam-

ples, the BT-based BNRM achieves accuracies of 74.2%, 83.6%, and 75.2% on Unified-Feedback, HHH Alignment, and MT-Bench, respectively, corresponding to improvements of 5.4%, 13.3%, and 6.1% points over the corresponding BT baseline. Likewise, the GRM-based BNRM attains 74.1%, 82.4%, and 75.1% on the same benchmarks, improving over GRM-SFT by 2.6%, 3.7%, and 2.1% points. In RewardBench, both BT-BNRM and GRM-BNRM achieve significant improvement over strong baselines under both 40/400K training splits, respectively. Table 1 indicates that our Bayesian non-negative framework effectively helps reward model suppresses reliance on spurious features, thereby improving generalization. Additional experiments on label noise setting, influence analysis of $\eta$, and convergence curve can be found in Appendix A.6.

**Full Parameter Training Results On the SP.** Skywork-Preference-v0.2 (SP) provides higher-quality preference data compared with UF dataset. We further refine the already strong Skywork-Reward-Llama-3.1-8B model and give training details in the Appendix C. Table 2 presents a comprehensive RewardBench evaluation, where our reward model achieves an overall score of 93.6 and reaches 89.7% and 92.6% on the Chat-hard and Safety subsets, respectively. In conclusion, these results show that our Bayesian non-negative reward model not only exhibits strong standalone preference modeling performance but also acts as a "plug-and-play" module that further enhances the generalization of existing powerful reward models.

**Advantages in Low-Resource and Noisy Settings.** A robust reward model should generalize well despite limited

*Table 3.* We adopt the official evaluation implementation of the `Evalscope` package by using 0-Shot, except for GSM8K, Race, and TriviaQA. The best result (accuracy %) in each column is in **bold**, and the second best is underlined.

| Benchmark | Llama3.1-8B-Instruct | | | | | | | OpenRLHF-Llama3-8B-SFT | | | | | | |
|---|---|---|---|---|---|---|---|---|---|---|---|---|---|---|
| | Base | SK | PoE | LP | ALBM | InfoRM | Ours | Base | SK | PoE | LP | ALBM | InfoRM | Ours |
| GSM8K$_{4shots}$ | 83.93 | **84.61** | 83.62 | 75.97 | 84.08 | 83.78 | 82.49$^{\downarrow1.44}$ | 74.83 | 78.17 | 77.79 | 77.18 | **78.85** | 76.74 | 77.10$^{\uparrow2.27}$ |
| Hellaswag | **77.21** | 76.42 | 77.08 | 73.15 | **77.21** | 76.78 | 74.66$^{\downarrow2.55}$ | 72.51 | **74.76** | 72.51 | 72.51 | 74.63 | 72.12 | 60.68$^{\downarrow11.83}$ |
| IFEval | 72.83 | 70.06 | 71.72 | 65.47 | 73.55 | 74.12 | **78.20**$^{\uparrow5.37}$ | 44.92 | 45.10 | 49.72 | 46.21 | 46.21 | 46.21 | **52.41**$^{\uparrow7.49}$ |
| MMLU | 72.31 | 72.33 | 71.97 | 65.13 | **72.57** | 72.22 | 70.72$^{\downarrow1.59}$ | 54.45 | 52.40 | 54.77 | 54.45 | 55.25 | 54.97 | **57.51**$^{\uparrow3.06}$ |
| Race$_{3shots}$ | 66.50 | 53.89 | 60.03 | 78.90 | 59.00 | 65.20 | **83.31**$^{\uparrow16.81}$ | 79.21 | 78.82 | **81.39** | 80.30 | 80.69 | 78.72 | 75.68$^{\downarrow3.53}$ |
| BBH | 64.52 | 65.69 | 60.50 | 61.10 | 64.84 | 66.13 | **67.72**$^{\uparrow3.20}$ | 61.20 | 62.68 | **62.69** | 62.28 | 61.10 | 61.62 | 55.71$^{\downarrow5.49}$ |
| HumanEval | 70.12 | 68.29 | 66.46 | 60.37 | 65.85 | 70.12 | **70.73**$^{\uparrow0.61}$ | 60.98 | 57.32 | 59.76 | 59.76 | 60.37 | 57.32 | **64.63**$^{\uparrow3.65}$ |
| TriviaQA$_{5shots}$ | 32.64 | 49.01 | 48.41 | 47.20 | 52.09 | 30.56 | **71.99**$^{\uparrow39.35}$ | 48.53 | 52.86 | 52.34 | 48.32 | 51.52 | 48.16 | **54.30**$^{\uparrow5.77}$ |
| Avg. Accuracy | 62.83 | 63.31 | 63.14 | 61.36 | 63.92 | 62.80 | **74.98**$^{\uparrow12.15}$ | 55.68 | 56.94 | 57.85 | 56.54 | 57.72 | 55.34 | **62.25**$^{\uparrow6.57}$ |

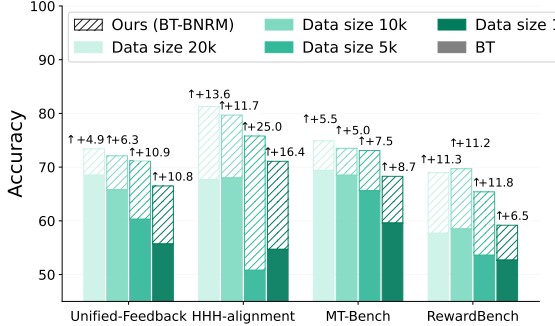

*(a)* Low-resource settings

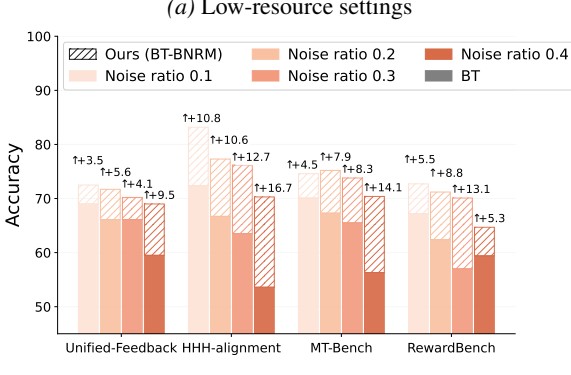

*(b)* Noisy settings

*Figure 4.* ID and OOD evaluation results for BT-BNRM. (a): performance when training on a varying number of samples. (b): performance under different label-noise ratios. Solid bars denote the BT baseline, hatched bars denote our BT-BNRM.

annotations and label noise—both common in practice. We evaluate BT-BNRM using a Gemma-2B-it backbone under two challenging settings: (1) Low-resource: training on 1K to 20K UF samples; (2) Label noise: training on 40K samples with noise rates from 0.1 to 0.4. Figure 4a shows that BNRM consistently outperforms BT with the performance gap widening as data volume decreases. Remarkably, BNRM trained on only 1K examples matches the performance of BT trained on 20K on RewardBench, with similar trends across other datasets. Under label noise (Figure 4b), BNRM demonstrates even greater resilience. At a 40% rate, BNRM improves BT by up to 16.7% and rivals

the performance of BT trained with only 10%-20% noise. These results show that our BNRM is both data-efficient and noise-tolerant, making it highly suitable for real-world scenarios with scarce high-quality preference data.

### 5.3. RLHF Evaluation

To further investigate whether BNRM can effectively mitigate reward hacking in real RLHF, we fine-tune Llama-3.1-8B-Instruct and OpenRLHF-Llama3-8B-SFT based on LoRA using Proximal Policy Optimization (PPO) (Schulman et al., 2017) on 20K samples from the alpaca-gpt4-data-en dataset (Peng et al., 2023). We use BNBT-Reward-Llama-3.1-8B fully fine-tuned on the SP dataset, and optimized policies are evaluated on widely used benchmarks, where we report Accuracy (%). As shown in Table 3, the two BNRM-fine-tuned policies achieve higher performance than their respective baselines, reaching 74.98% and 62.25%, indicating that BNRM can effectively simulate human preferences to guide LLM fine-tuning toward better performance. Additional training and evaluation details can be found in Appendix C. Further, we conduct the Best-of-N (BoN) test in Appendix D, which shows that the BT-based RM suffers from reward hacking as KL increases, while our BNRM remains aligned with the gold reward.

**Arena-Hard and Human Assessment.** To examine whether the mitigation of reward hacking leads to better alignment with human preferences rather than merely better agreement with another reward model, we provide a two-fold validation using both high-correlation benchmarks and direct human assessment. (1) We use Arena-Hard-v0.1 benchmark, specifically designed to yield the highest correlation with the Chatbot Arena (Li et al., 2024). Using GPT-4.1 as a judge, our BNRM-aligned Llama-3.1-8B-Instruct achieves a 50% win rate and 28% tie rate against the base Llama-3.1-8B-Instruct, significantly outperforming the baseline (22% win rate). (2) To provide direct evidence, we conducted a blind pairwise human evaluation. Two experts with PhD backgrounds evaluated 50 randomly sampled pairs from Arena-Hard. Our model

*Table 4.* RLHF alignment results on Arena-Hard-v0.1 for the PPO-tuned BNRM policy model based on Llama-3.1-8B-Instruct. The results are evaluated by GPT-4.1 and two human experts. All win, tie, and lose rates are reported from the BNRM side.

| Evaluator | Opponent | Win | Tie | Lose |
|---|---|---|---|---|
| GPT-4.1 | Llama-3.1-8B-Instruct | 0.4989 | 0.2772 | 0.2239 |
| GPT-4.1 | Mistral-7B-Instruct | 0.5850 | 0.2045 | 0.2105 |
| Human Evaluator 1 | Llama-3.1-8B-Instruct | 0.5200 | 0.0600 | 0.4200 |
| Human Evaluator 2 | Llama-3.1-8B-Instruct | 0.5000 | 0.3000 | 0.2000 |
| Human Average | Llama-3.1-8B-Instruct | 0.5100 | 0.1800 | 0.3100 |

*Table 5.* Performance of different reward models on RM-Bench. The best result in each column is in **bold**, and the second best is underlined. BT/GRM-BNRM are based on BT/GRM-SFT.

| Model | Total | Chat | Math | Code | Safety | Hard | Normal | Easy |
|---|---|---|---|---|---|---|---|---|
| BT | 57.3 | 47.7 | 52.2 | **54.0** | 75.1 | 33.6 | 61.7 | 76.4 |
| BT-Margin | 56.8 | **52.5** | 52.3 | 51.9 | 70.6 | 34.9 | 60.7 | 74.9 |
| GRM-DPO | 56.8 | 49.5 | 52.4 | 50.0 | 75.1 | 30.2 | 60.3 | 79.8 |
| GRM-DPO w/o ref | 57.0 | 49.0 | 52.8 | 50.3 | 76.1 | 33.5 | 60.8 | 76.9 |
| GRM-SFT | 57.1 | 49.1 | 52.8 | 50.3 | 76.0 | 32.0 | 60.5 | 78.6 |
| BT-BNRM | **60.4**$^{\uparrow 3.1}$ | 50.5$^{\uparrow 2.8}$ | **55.2**$^{\uparrow 3.0}$ | 49.5$^{\downarrow 4.5}$ | **86.5**$^{\uparrow 11.4}$ | **36.3**$^{\uparrow 2.7}$ | **62.1**$^{\uparrow 0.4}$ | **82.9**$^{\uparrow 6.5}$ |
| GRM-BNRM | 59.2$^{\uparrow 2.1}$ | 48.3$^{\downarrow 0.8}$ | **55.2**$^{\uparrow 2.4}$ | 49.1$^{\downarrow 1.2}$ | 84.0$^{\uparrow 8.0}$ | 34.0$^{\uparrow 2.0}$ | 61.1$^{\uparrow 0.6}$ | 82.3$^{\uparrow 3.7}$ |

achieved a 51% win rate and 18% tie rate, consistently mirroring the trends observed in our LLM-as-a-judge results. These consistent gains across both automated and human rubrics demonstrate that BNRM's mitigation of reward hacking translates into genuine improvements in response quality, rather than merely overfitting to another reward model. More detailed results are reported in Table 4

### 5.4. Reward Debiasing and Interpretability

Despite being proposed without explicit debiasing supervision, BNRM possesses an inherent theoretical capacity to mitigate common reward biases. This section empirically validates debiasing capability by focusing on length and formatting biases. Beyond robust performance, we find that the global factors $\Phi$ can effectively rectify specific preference errors made by the local factors $\theta$, which brings a mechanistic interpretability that is typically unattainable in conventional scalar-based reward models. Unless otherwise stated, all reward models are LoRA fine-tuned on Gemma-2B-it using a 40K subset of the training data.

**Length Debiasing and Format Debiasing.** We evaluate the debiasing capabilities of reward models using the RM-Bench. Specifically, the RM-Bench Hard subset contains samples where rejected responses are deliberately crafted to be longer and better-formatted than the preferred ones. We leverage this subset to quantify the sensitivity of reward scores to response length using the Pearson correlation coefficient (Benesty et al., 2009). As shown in Table 5, BNRM significantly outperforms the baselines on the Hard subset while maintaining robust performance across other categories. Figure 5 illustrates the behavior of various RMs under length bias without any explicit debiasing supervision. The vanilla BT model exhibits a high Pearson correlation ($r = 0.488$), indicating a strong spurious correlation between response length and perceived quality. In contrast, BNRM achieves a substantially lower correlation of 0.123, outperforming all strong baselines. This suggests that BNRM effectively mitigates reliance on surface heuristics like length or formatting, instead guiding the model to capture intrinsic, fine-grained preference signals. More detailed results and Pearson analyses for additional methods

are provided in Appendix B.

**Sparsity, Non-negativity, and Interpretability.** To better investigate how the sparsity and non-negativity of our framework effectively mitigate reward hacking and response biases, we visualize a subset of the $\theta$ and $\Phi$ factors in Figure 6, revealing two primary mechanisms through which BNRM operates: (1) Signal Amplification: When $\theta$ correctly captures the preference signal (*i.e.*, the chosen response exhibits higher activations than the rejected one on factors such as F-433, F-491, and F-238), the global factors $\Phi$ further enlarge the preference margin. (2) Error Rectification: Conversely, for factors like F-455, F-189, and F-890 where $\theta$ alone would mis-rank the pair, the sparsity of $\Phi$ effectively suppresses these erroneous signals by driving their global weights toward zero. In our analysis, these two scenarios occur in 1,936 and 761 samples, accounting for 64.9% and 25.5% of the test set, respectively. This provides strong empirical evidence that BNRM serves as a robust proxy for true human preferences under noisy and data-limited conditions, while offering *mechanistic interpretability* typically unattainable by previous scalar-based reward models. Furthermore, we employ GPT-5 to perform semantic analysis on the top-$k$ factors identified by their $\Phi$ weights and their associated responses. The specific prompts are detailed in Table 10, and the resulting factor-level semantics are summarized in Table 11 of Appendix B.

## 6. Conclusion

In this paper, we proposed Bayesian Non-Negative Reward Model (BNRM), which integrates the interpretability of non-negative factor analysis with the scalability of large language models. By introducing a Weibull-parameterized amortized inference network, BNRM achieves efficient posterior inference with minimal parameter overhead, while providing principled uncertainty estimates. Empirical results demonstrate that BNRM effectively mitigates reward overoptimization, improves robustness under distributional shift, and yields more interpretable reward decompositions than ensemble-based or purely neural baselines. These findings suggest that probabilistic sparsity and uncertainty modeling can serve as powerful inductive biases for building reliable reward models.

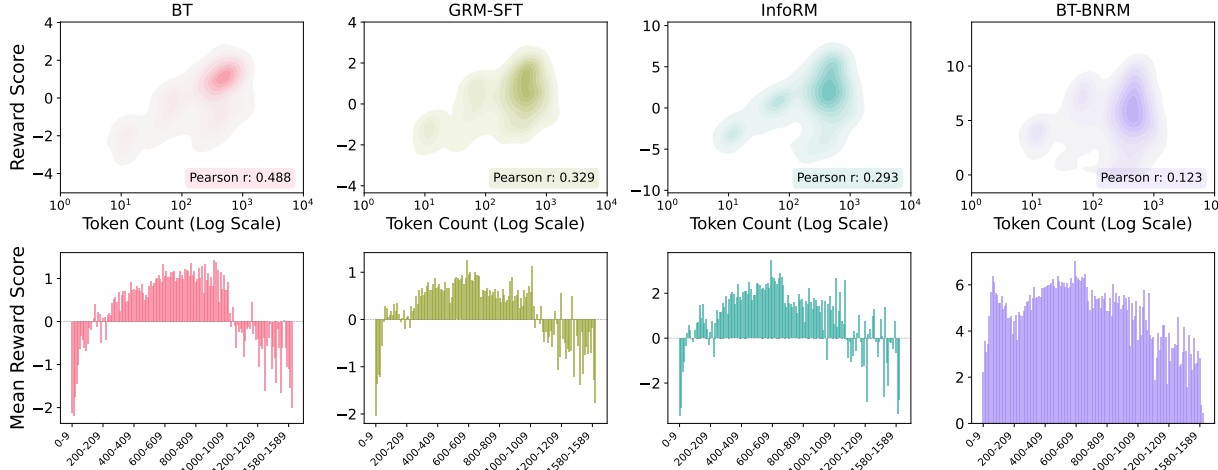

*Figure 5.* Pearson correlation and mean reward score between response length and reward score on the RM-Bench Hard subset. The top plot shows the correlation between response length and reward score. The x-axis is log-scaled for better visual clarity. The bottom plot reports the average reward score within each length bucket, which visually highlights the non-negative property of our BT-BNRM.

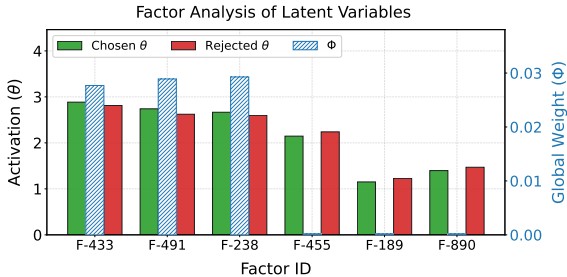

*Figure 6.* Partial $\theta$ and $\Phi$ factor activations for the chosen and rejected responses on RewardBench. The left y-axis denotes the activation strength of $\theta$, and the right y-axis denotes the activation strength of $\Phi$, which represents the global weight of each latent factor in $\theta$. Bars with $\Phi$ values close to zero correspond to factors that are effectively inactive.

**Limitations.** Although BNRM improves reward modeling robustness through Bayesian non-negative factorization, several limitations remain. First, BNRM introduces an additional method-specific hyperparameter, the KL coefficient $\eta$, which controls the strength of Bayesian regularization between the variational posterior and the prior. Although BNRM remains robust under extreme choices of $\eta$ and still outperforms the strong baseline across all datasets, achieving the best overall performance may still require tuning $\eta$ to balance prior regularization and posterior adaptation. In our experiments, $\eta = 10^{-5}$ provides a favorable balance. Second, although our OOD, noisy-label, Arena-Hard, and human evaluation results provide evidence that BNRM mitigates reward hacking beyond proxy-based reward scores, open-ended RLHF settings may expose more diverse and adaptive forms of reward hacking. Evaluating BNRM under broader multi-turn, tool-use, and long-horizon alignment scenarios remains an important direction for future work.

## Acknowledgements

The authors would like to thank the anonymous reviewers for their valuable comments and constructive suggestions. This work of G. Rong and D. Guo was supported by the National Natural Science Foundation of China (NSFC) under Grant (No. 62306125). This work of Z. Duan, B. Chen was supported in part by the National Natural Science Foundation of China under Grant 62576266 and U21B2006; in part by the Fundamental Research Funds for the Central Universities QTZX24003 and QTZX23018; in part by the 111 Project under Grant B18039. Part of this work was carried out during Dandan Guo's visit to King Abdullah University of Science and Technology (KAUST).

## Impact Statement

This paper presents work whose goal is to advance the field of Machine Learning. There are many potential societal consequences of our work. This work aims to improve the robustness and interpretability of reward modeling for RLHF by mitigating reward hacking and reducing reliance on spurious preference signals. These improvements can support safer and more reliable alignment of large language models, especially in settings where reward models are exposed to noisy, biased, or distribution-shifted preference data. At the same time, more robust reward optimization techniques are dual-use. If the reward objective is misspecified or intentionally harmful, methods that make optimization more effective could also be used to improve models toward undesirable or malicious goals. Therefore, BNRM should be applied with careful reward design, data auditing, human oversight, and downstream safety evaluation, particularly in open-ended deployment settings.

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

# A. Method Analysis

## A.1. Algorithm

Algorithm 1 details the BNRM pipeline, which takes preference data as input, computes the latent variables $\theta$ and $\Phi$, and then uses the resulting reward scores, together with the BT preference loss, to fine-tune pre-trained LLM. The variational inference network employs two separate MLPs to extract features and compute the shape and scale hyperparameters for reparameterized sampling from the Weibull distribution, where each MLP is a single-layer network with 1024 hidden neurons.

## A.2. Complexity Analysis

BNRM introduces only a lightweight modification to the standard reward modeling pipeline, incurring negligible additional space and computational overhead. As in conventional preference-based reward models, the dominant cost arises from the feature extractor (*e.g.*, a Transformer or pre-trained LLM), with complexity

$$O\left(L \times \left(N^2 \times d_{\text{model}} + N \times d_{\text{model}}^2\right)\right), \quad (10)$$

where $L$ denotes the number of layers in the feature extractor, $N$ is the input sequence length, and $d_{\text{model}}$ is the hidden dimension. Within BNRM, the additional cost arises from computing the KL divergence terms associated with the sparse non-negative latent variables $\boldsymbol{\theta}$ and the global Bayesian parameters $\Phi$. Specifically, the complexity of local latent variable updates is $O(K)$, while that of global latent variable updates is $O(K \times 1)$, where $K$ is the number of latent factors. Since $K \ll d_{\text{model}} \times N$, these costs are negligible relative to the overall feature extraction. Unlike ensemble-based uncertainty estimation, which requires training and storing multiple full reward models, BNRM introduces only a small number of additional parameters via sparse latent variables and a lightweight variational inference network.

## A.3. Reproducibility of Factorization and Open-Ended RLHF Settings

**Reproducibility of Factorization.** The reward in BNRM is constructed as $r(x, y) = \theta_v^\top \Phi$, where $\theta_v$ denotes the instance-specific latent factor and $\Phi$ denotes the global reward dictionary. This formulation can be viewed as a constrained non-negative matrix factorization problem, $\min_{\theta \geq 0, \Phi \geq 0} \left\| r - \theta^\top \Phi \right\|_F^2$. Following the partial identifiability analysis of non-negative matrix factorization (Gillis & Rajkó, 2023), such factorizations are partially identifiable under two strict conditions: (a) a selective window condition, where there exists a row $j$ such that $\Phi(j, :) = \alpha e_{(k)}^\top$ for some $\alpha > 0$, meaning that a specific factor is uniquely activated by a distinct pattern, and (b) a

sparsity constraint. BNRM naturally satisfies both conditions through its non-negative architecture and Weibull-parameterized sparse priors. This mathematically guarantees that the decomposition is unique up to permutation and scaling, ensuring that the learned factors correspond to distinct structural properties in the preference data.

We further conduct empirical stability tests across multiple random seeds, model scales, and datasets, with detailed results reported in Table 7. While exact factor indices may permute, BNRM robustly recovers seven core semantic families across all settings, such as safety refusal and concise problem-solving. For instance, the safety refusal concept, which corresponds to Factor 343 in our main run, consistently emerges with the same behavioral triggers in other seeds, matching Factors 205, 823, and 917. This cross-run consistency is anchored by the global dictionary $\Phi$, which acts as a population-level regularizer to suppress noise and ensure reproducible disentangled representations.

**Implementation Complexity and Training Stability of Variational Inference.** Having discussed the reproducibility and semantic stability of the sparse factors constructed by BNRM, we further examine whether introducing variational inference and Weibull reparameterization leads to additional implementation complexity or training instability. *For Implementation Complexity,* BNRM introduces additional structure compared to deterministic BT, but this overhead is remarkably lightweight. As discussed in Appendix A.2, we provide a detailed complexity analysis, where the additional computational overhead introduced by BNRM is limited relative to the backbone. To further validate this efficiency, we report the wall-clock training time for full-parameter fine-tuning on the Skywork-Reward-Preference-v0.2 dataset in Table 6. For the Gemma-2B-it model, BNRM increases training time by only 7.7%, from 3.88h to 4.18h. For the larger Llama-3.1-8B-Instruct model, the overhead is even more minimal, representing only a 1.3% increase, from 11.70h to 11.86h. This trend indicates that as the model scale increases, the marginal cost of BNRM becomes nearly imperceptible, making it suitable for modern large-scale RLHF workflows. *For Training Stability,* The Weibull distribution is specifically chosen for its ability to model sparse and positive latent variables while admitting an efficient and stable reparameterization path. Despite the use of variational inference and Weibull reparameterization, BNRM still exhibits training stability. Our localized uncertainty modeling, which confines stochasticity to the output factors, prevents training instability from propagating through the backbone. As shown in Figure 8, BNRM achieves a validation accuracy of 71.75% within only 0.25 epochs, a level that standard BT and GRM variants struggle to reach even after 1.5 epochs. Throughout our experiments, BNRM converges

---

**Algorithm 1** Reward Modeling with Non-negative Bayesian

---

1: **Input:** Preference dataset $\mathcal{D} = \{(\boldsymbol{x}_i, \boldsymbol{y}_i^1, \boldsymbol{y}_i^2)\}_{i=1}^N$, KL Divergence coefficient $\eta$.
2: **Output:** Trained reward model $R$.
3: Initialize backbone parameters $W_{llm}$ and Non-negative Bayesian Head parameters $(\text{MLP}_\ell, \text{MLP}_k, \text{MLP}_{kw}, W, b)$.
4: **while** not converged **do**
5:   Sample mini-batch $\{(x_i, \boldsymbol{y}_i^1, \boldsymbol{y}_i^2)\}_{i=1}^B \sim \mathcal{D}$.
6:   Construct $2B$ sequences $\{(x_i, y_i^1)\}_{i=1}^B$ and $\{(x_i, y_i^2)\}_{i=1}^B$.
7:   Extract representations: $H \in \mathbb{R}^{2B \times L \times d} \leftarrow \text{W}_{\text{llm}}(\text{mini-batch})$.
8:   **Local Sparse Representation** $\theta$
9:   Build feature of inference net $z_{\text{out}} \in \mathbb{R}_+^{2B \times L \times 1024} \leftarrow \text{ReLU}(\text{MLP}_\ell(H))$.
10:   Build Weibull parameter: $\kappa_\theta \in \mathbb{R}_+^{2B \times L \times 1024} \leftarrow 1 + \text{softplus}(\text{MLP}_k(H))$.
11:   Build Weibull parameter: $\lambda_\theta \leftarrow z_{\text{out}}/\exp\big(\Gamma(1 + 1/\kappa_\theta)\big)$.
12:   Sample noise $u_\theta \sim \text{Uniform}(0, 1)^{\text{shape}(\lambda_\theta)}$.
13:   Reparameterized sampling: $\theta \in \mathbb{R}_+^{2B \times L \times 1024} \leftarrow \lambda_\theta \cdot \big(-\log(1 - u_\theta)\big)^{1/\kappa_\theta}$.
14:   KL regularization: $\text{KL}_\theta \leftarrow \text{KL}(\text{Gamma}(1, 1) \| \text{Weibull}(\kappa_\theta, \lambda_\theta))$.
15:   **Global Reward Dictionary** $\Phi$
16:   Build feature of inference net $z_{\text{out}}^{(w)} \in \mathbb{R}_+^{1024 \times 1} \leftarrow \text{ReLU}(W^\top)$.
17:   Build Weibull parameter: $\kappa_\Phi \leftarrow 1 + \text{softplus}(\text{MLP}_{kw}(z_{\text{out}}^{(w)}))$.
18:   Build Weibull parameter: $\lambda_\Phi \leftarrow z_{\text{out}}^{(w)}/\exp\big(\Gamma(1 + 1/\kappa_\Phi)\big)$.
19:   Sample noise: $u_\Phi \sim \text{Uniform}(0, 1)^{\text{shape}(\lambda_\Phi)}$.
20:   Reparameterized sampling: $\Phi \in \mathbb{R}_+^{1024 \times 1} \leftarrow \lambda_\Phi \cdot \big(-\log(1 - u_\Phi)\big)^{1/\kappa_\Phi}$.
21:   KL regularization: $\text{KL}_\Phi \leftarrow \text{KL\_GamWei}(\text{Gamma}(1, 1) \| \text{Weibull}(\kappa_\Phi, \lambda_\Phi))$.
22:   Calculate Reward Score $r \in \mathbb{R}^{2B \times L \times 1} \leftarrow \theta \cdot \Phi + \text{ReLU}(b)$.
23:   $\mathcal{L} \leftarrow -\frac{1}{B} \sum_{i=1}^B \log \sigma(r_i^1 - r_i^2) + \eta \cdot (\text{KL}_\theta + \text{KL}_\Phi)$.
24:   Update all parameters by one gradient step on $\mathcal{L}$.
25: **end while**

---

*Table 6.* Comparison of training time between BNRM and BT under full-parameter fine-tuning on Skywork-Reward-Preference-80K-v0.2.

| Method | Gemma-2B-it | Llama-3.1-8B-Instruct |
|--------|-------------|-----------------------|
| BT | 3.88h | 11.70h |
| BNRM | 4.18h | 11.86h |

to a higher and more stable validation plateau. This suggests that, rather than introducing optimization difficulty, the non-negative sparsity constraints act as an effective regularizer, filtering out biased preference signals and leading to a more reliable optimization trajectory.

**Larger-Scale and Open-Ended RLHF Settings.** Scaling RLHF to open-ended settings introduces more complex reward hacking challenges. (a) From the computational perspective, scaling any RLHF pipeline inherently demands significant resources for the LLM backbone, while the marginal barrier introduced by BNRM is negligible. As shown in our computational analysis A.2, the Bayesian layer adds only a 1.3% training time overhead on an 8B model. Thus, BNRM scales alongside standard reward models without imposing new computational bottle-

necks. (b) From the algorithmic perspective, the increased complexity of reward hacking at scale is precisely where BNRM provides value. Under strong optimization pressure, standard dense reward heads are more vulnerable to proxy misspecification because they may conflate genuine intent with complex shortcuts. In contrast, BNRM's sparse factorized formulation $r = \theta^\top \Phi$, disentangles these signals and tightens the generalization bound under distribution shifts. This design improves resistance to overoptimization in complex and open-ended trajectories, as empirically supported by our OOD, noisy-label, and Arena-Hard results.

### A.4. Comparison with Uncertainty-Aware Reward Models.

Among our baselines, InfoRM is the primary uncertainty-aware reward modeling method. However, BNRM provides a fundamentally different approach to uncertainty modeling. While InfoRM mitigates overoptimization indirectly through a variational information bottleneck and latent outlier signals, BNRM explicitly models uncertainty within a structured reward decomposition $r(x, y) = \theta^\top \Phi$. By applying a Bayesian treatment to both the instance-level factors $\theta$ and the global dictionary $\Phi$, BNRM goes

*Table 7.* Comparison of factor interpretability between the main-experiment BNRM and variants across different runs, a different model, and a different dataset. The different model setting uses Gemma2-2B-it, and the different dataset setting uses Skywork-Preference-v0.2 (SP).

| Semantic Family | Main-exp. BNRM | Different Runs-1 | Different Runs-2 | Different Model | Different Dataset |
|---|---|---|---|---|---|
| Prompt repetition / marker leakage | 236, 55, 421 | – | 231, 490, 746, 959 | 285 | 1 |
| Overly shallow responses / one-line code / incomplete implementation | 238, 586, 545, 443 | 264, 737, 821, 884 | 128, 396 | 146, 456 | 65, 1020 |
| Step-by-step guidance / list-style / template-like how-to | 493, 491 | 370 | 182 | 146 | 587, 958, 972 |
| Safety refusal / safety-domain behavior / polite alternative suggestions | 343, 433 | 205, 823, 917 | 396, 966 | 428, 749 | 65, 662, 958 |
| Mathematical / formal / concise problem-solving style | 918 | 536, 821 | – | – | 856 |
| Low-quality / meaningless / chaotic outputs | 2 | – | 93, 839 | 285 | 1 |
| Unsafe direct compliance / harmful enumeration / unsafe compliance | 691 | 823 | 411, 422 | 146, 456 | 662 |

beyond simply penalizing uncertainty and provides a disentangled and interpretable representation that prevents the model from becoming overconfident on specific spurious features. This structural advantage translates into consistent empirical gains across multiple dimensions. In direct reward modeling, BNRM consistently outperforms InfoRM on both ID and OOD evaluations, including UF, HHH, MTBench, and RewardBench, as shown in Table 1. This superiority further extends to downstream policy alignment, where BNRM achieves stronger RLHF performance on both Llama-3.1-8B-Instruct and OpenRLHF-Llama3-8B-SFT, as shown in Table 3. Finally, the proposed formulation yields stronger unsupervised length debiasing, as confirmed in Figure 5. Together, these results indicate that BNRM provides a more interpretable formulation and stronger practical robustness than existing uncertainty-aware baselines.

## A.5. Capacity for Modeling Complex Preference Interactions.

BNRM operates through a synergy between a highly non-linear LLM backbone and a structured Bayesian output layer. While the final reward aggregation $r = \theta^\top \Phi$ is a linear dot product, the mapping from the raw input $(x, y)$ to the local latent variables $\theta$ is performed by the deep backbone, which is fully capable of capturing intricate and non-monotonic preference interactions. By enforcing non-negativity and sparsity only at the final layer, BNRM ensures that the captured signals remain human-interpretable without sacrificing the underlying feature extraction power of the model.

BNRM also maintains structural parity with standard BT reward models, which typically project dense hidden representations $z$ to a scalar reward through a single linear head $W_{bt}$. By reconfiguring this projection into a factorized form, BNRM does not fundamentally restrict expressiveness compared with mainstream reward models, but instead introduces a principled inductive bias that enhances robustness. This architectural choice is also supported by the Linear Representation Hypothesis (Park et al., 2024), which suggests that high-level semantic concepts are often approximately linearly encoded in LLM latent spaces.

Consequently, a structured linear head at the output stage is sufficient to recover these concepts while providing the added benefit of disentanglement.

Our empirical findings further validate that this factorized formulation preserves sufficient expressivity while offering stronger generalization. As shown in Table 1 and Figure 4, BNRM consistently outperforms the dense BT baseline, including complex reasoning and chat-hard tasks. The performance gap remains substantial even under 40% label noise, indicating that the non-negative factorization acts as a robust regularizer that filters out spurious shortcuts without compromising the model's ability to capture genuine human intent.

## A.6. Further Experiments

**Robustness to Label Noise.** Across ID, OOD, and RewardBench evaluations with 25% label noise, BNRM consistently outperforms classical BT and GRM baselines. These gains are especially pronounced in Safety and Reasoning, where noisy supervision often amplifies spurious correlations. Importantly, while larger models and more data improve overall accuracy, BNRM's probabilistic sparsity and uncertainty modeling yield robust performance even under corrupted labels, a setting unavoidable in real-world preference collection. As shown in Tables 8, we further train BNRM with LoRA on two base models using 40/400K examples sampled from the UF dataset. With 40K training data, our method improves ID performance over the BT baseline by 5.8% and 3.4% on the two models, and achieves gains of 6.5% and 13.1% on RewardBench, while also significantly outperforming GRM-SFT. With 400K training data, BNRM still yields improvements of 3.2% and 3.1% on RewardBench across the two models, consistently surpassing all baselines and prior effective methods. This demonstrates that BNRM not only scales with data but also provides a principled safeguard against the noise and brittleness inherent in human feedback.

**Hyperparameter Sensitivity Analysis.** The previous {Influence Analysis of $\eta$} has been replaced. BNRM is robust and requires minimal tuning. In our implementation, all training settings except the KL coefficient $\eta$ follow GRM (Yang et al., 2024b), and $\eta$ is the only method-

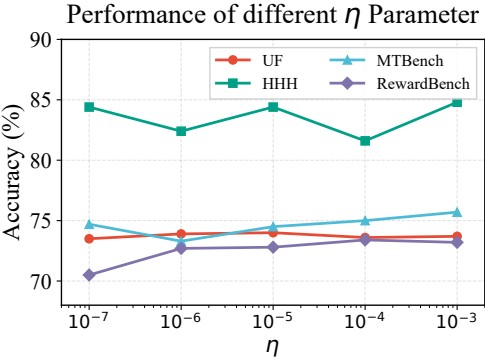

*Figure 7.* Influence of different values of $\eta$ on in-distribution (ID) and out-of-distribution (OOD) performance, where $\eta$ controls the strength of the KL-divergence regularization term relative to the BT preference loss.

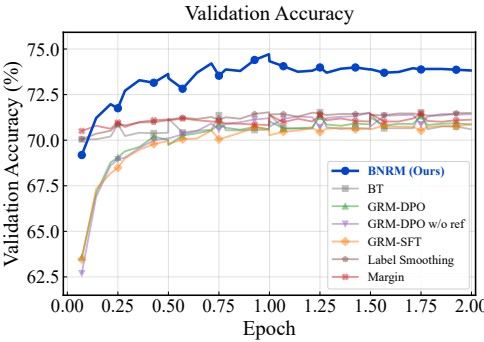

*Figure 8.* Validation accuracy on Unified-Feedback with 40K training examples, fine-tuning Gemma-2B-it with LoRA. Our BNRM consistently achieves higher validation accuracy throughout training and at convergence compared with BT, GRM-DPO, GRM-DPO w/o ref, GRM-SFT, Label-smooth, and Margin.

specific hyperparameter introduced by the Bayesian non-negative reward formulation defined in Eq. 9. To examine how different KL regularization strengths affect reward model performance, we train BT-BNRM on Gemma-2B-it with 40K preference pairs sampled from the UF dataset using LoRA. As shown in Figure 7, decreasing $\eta$ weakens the KL-divergence regularization term, allowing the variational posterior to deviate more from the prior, while increasing $\eta$ enforces a stronger match. Both extremes adversely affect the preference learning of BNRM and lead to degraded performance on all four datasets. Therefore, $\eta = 10^{-5}$ achieves the highest overall average accuracy, indicating that a well-balanced $\eta$ is crucial for our Bayesian non-negative reward framework, as it effectively controls the latent space and thereby improves model performance. Nevertheless, BNRM remains reasonably robust across a wide range of values, and the performance variation across $\eta$ remains moderate. Crucially, even under extreme choices of $\eta$, BNRM still retains stronger reward-modeling performance than the BT baseline across all datasets, while $\eta = 10^{-5}$ provides a favorable balance between prior regularization and posterior adaptation. Even at $\eta = 10^{-6}$, which corresponds to the weakest-performing setting in our experiments, BNRM still substantially outperforms BT, improving RewardBench accuracy from 64.5% to 72.7%. These results suggest that BNRM benefits from a balanced KL coefficient but does not require substantial hyperparameter tuning to work well.

**Convergence and Validation Accuracy Comparison.** We compare the convergence behavior and validation performance of different methods under the same training conditions. Concretely, we fine-tune Gemma-2B-it with LoRA on 40K training examples sampled from Unified-Feedback. Figure 8 plots validation accuracy against training epochs for all baselines and our BNRM. We observe that BNRM surpasses a validation accuracy of 71.75% as early as 0.25 epochs, outperforming BT, GRM, and their variants,

whereas the baselines struggle to reach this level even after 1.5 epochs. BNRM eventually converges around 74% validation accuracy. Overall, these results indicate that, under the same data and compute budget, BNRM does not introduce additional optimization difficulty. Instead, it converges to a higher and more stable validation performance, suggesting that its sparsity constraints help filter out biased or spurious preference signals.

## B. Length and Formatting Debiasing and Interpretability

This section provides additional results complementing Section 5.4. Figure 9 reports the correlation between response length and proxy reward scores on the RM-Bench Hard subset for more baseline methods, together with our GRM-BNRM variant. We observe that, across Margin, GRM-DPO, and GRM-DPO w/o ref, GRM-BNRM consistently attains lower correlations than their respective BT/GRM counterparts. This further supports that the sparsity and non-negativity in BNRM act as an effective regularizer against reward hacking and bias, encouraging the reward model to focus on the actual content quality of responses rather than superficial length or formatting cues. Table 5 presents the detailed performance of BNRM and several strong baseline methods. BNRM's two variants still achieve 60.4% and 59.2% accuracy, and obtain gains of 2.7 and 2.0 percentage points on the Hard subset. These results provide concrete evidence that BNRM maintains its advantage even when superficial cues are deliberately confounded with response quality.

We use the prompts listed in Table 10 and employ GPT-5 to assist our semantic analysis. Table 11 reports three representative factor semantics, which were selected from $\Phi$ via a top-k (k=20), and which, in turn, correspond to strong activations of the associated factors in $\theta$ under negative, mixed, and positive response contexts, respectively.

*Table 8.* Results on ID and OOD evaluation with 40/400K Unified-Feedback training examples under 25% **label noise** using LoRA. Best is **bold** and second-best is underlined. Superscripts indicate the performance change of our methods, including BT-BNRM vs. BT and GRM-BNRM vs. GRM-SFT.

| Reward Model | Gemma 2B it | | | | | | | | Gemma2 2B it | | | | | | | |
|---|---|---|---|---|---|---|---|---|---|---|---|---|---|---|---|---|
| | Unified Feedback | HHH Alignment | MT Bench | RewardBench | | | | | Unified Feedback | HHH Alignment | MT Bench | RewardBench | | | | |
| | | | | Average | Chat | Chat-Hard | Safety | Reasoning | | | | Average | Chat | Chat-Hard | Safety | Reasoning |
| *Unified-Feedback 40K* | | | | | | | | | | | | | | | | |
| BT | 66.0 | 61.5 | 65.8 | 61.1 | 86.3 | **43.9** | 54.9 | 59.3 | 71.2 | 82.4 | 72.2 | 67.5 | 95.9 | 48.7 | 72.4 | 53.0 |
| BT-Margin | 68.5 | 67.7 | 69.2 | 62.1 | 93.8 | 40.1 | 56.4 | 58.1 | 74.8 | 80.7 | 74.6 | 73.9 | 97.2 | 46.1 | 79.5 | 72.8 |
| BT-LabelSmooth | 65.7 | 66.7 | 63.6 | 65.8 | 84.9 | 39.0 | 66.9 | 72.4 | 71.8 | 79.7 | 71.8 | 74.7 | 93.9 | 48.7 | 81.6 | 74.6 |
| GRM-SFT | 71.1 | 76.0 | **74.9** | 65.0 | 94.6 | 37.1 | 74.3 | 53.9 | 75.1 | 85.1 | 74.7 | 78.8 | **97.4** | 54.1 | **86.9** | 76.9 |
| BT-BNRM | $71.8^{\uparrow5.8}$ | $78.3^{\uparrow16.8}$ | $74.2^{\uparrow8.4}$ | $67.6^{\uparrow6.5}$ | $95.0^{\uparrow8.7}$ | $43.2^{\downarrow0.7}$ | $78.1^{\uparrow23.2}$ | $54.2^{\downarrow5.1}$ | $74.6^{\uparrow3.4}$ | $86.9^{\uparrow4.5}$ | $74.8^{\uparrow2.6}$ | $80.6^{\uparrow13.1}$ | $96.1^{\uparrow0.2}$ | $59.2^{\uparrow10.5}$ | $85.1^{\uparrow12.7}$ | $81.8^{\uparrow28.8}$ |
| GRM-BNRM | $71.8^{\uparrow0.7}$ | $77.4^{\uparrow1.4}$ | $74.8^{\downarrow0.1}$ | $66.4^{\uparrow1.4}$ | $95.0^{\uparrow0.4}$ | $40.1^{\uparrow3.0}$ | $77.7^{\uparrow3.4}$ | $52.6^{\downarrow1.3}$ | $75.7^{\uparrow0.6}$ | $86.4^{\uparrow1.3}$ | $76.0^{\uparrow1.3}$ | $80.1^{\uparrow1.3}$ | $96.8^{\downarrow0.6}$ | $57.3^{\uparrow3.2}$ | $85.9^{\downarrow1.0}$ | $80.4^{\uparrow3.5}$ |
| *Unified-Feedback 400K* | | | | | | | | | | | | | | | | |
| BT | 69.5 | 73.3 | 70.5 | 67.3 | 91.3 | 40.5 | 65.5 | **71.9** | 74.5 | 81.9 | 73.6 | 76.4 | 96.7 | 49.3 | 82.7 | 76.7 |
| BT-Margin | 71.9 | 77.6 | 73.3 | 69.4 | 95.5 | 39.7 | 71.5 | 70.7 | 76.0 | 84.9 | 74.8 | 75.4 | 98.0 | 48.9 | 81.1 | 73.7 |
| BT-LabelSmooth | 68.5 | 67.2 | 69.6 | 66.8 | 89.7 | 36.0 | 63.4 | 78.1 | 74.3 | 83.9 | 73.8 | 78.4 | 96.7 | 52.2 | 79.6 | 85.0 |
| GRM-SFT | 75.3 | 83.3 | 76.2 | 69.8 | 95.8 | 41.2 | **80.3** | 61.8 | 75.2 | 84.4 | 73.7 | 75.8 | 96.7 | 49.3 | 83.9 | 73.2 |
| BT-BNRM | $74.4^{\uparrow4.9}$ | $84.7^{\uparrow11.4}$ | $75.0^{\uparrow4.5}$ | $70.5^{\uparrow3.2}$ | $95.8^{\uparrow4.5}$ | $38.6^{\downarrow1.9}$ | $78.5^{\uparrow13.0}$ | $69.1^{\downarrow2.8}$ | $77.4^{\uparrow2.9}$ | $85.9^{\uparrow4.0}$ | $77.2^{\uparrow3.6}$ | $79.5^{\uparrow3.1}$ | $97.8^{\uparrow1.1}$ | $48.6^{\downarrow0.7}$ | $84.1^{\uparrow1.4}$ | $87.5^{\uparrow10.8}$ |
| GRM-BNRM | $75.1^{\downarrow0.2}$ | $82.1^{\downarrow1.2}$ | $75.6^{\downarrow0.6}$ | $71.4^{\uparrow1.6}$ | $96.0^{\uparrow0.2}$ | $41.7^{\uparrow0.5}$ | $78.9^{\uparrow1.4}$ | $68.8^{\uparrow7.0}$ | $77.7^{\uparrow2.5}$ | $87.0^{\uparrow2.6}$ | $75.5^{\uparrow1.8}$ | $77.5^{\uparrow1.7}$ | $97.1^{\uparrow0.4}$ | $51.3^{\uparrow2.0}$ | $84.9^{\uparrow1.0}$ | $76.5^{\uparrow3.3}$ |

*Figure 9.* Pearson correlation and mean reward score between response length and reward score on the RM-Bench Hard subset. The top plot shows how the correlation between response length and reward score. The x-axis is log-scaled for better visual clarity. The bottom plot reports the average reward score within each length bucket, which visually highlights the non-negative property of our BNRM.

# C. Bayesian Non-negative RM Training Details

In this section, we provide detailed descriptions of all experimental settings.

## C.1. Reward Modeling

Unless otherwise specified, all experiments are conducted with $\eta = 1e-5$ under the configuration in Eq. 9. We train the reward models using LoRA and full fine-tuning on the Unified Feedback[1] and SP datasets, respectively, and the detailed hyperparameter configurations used during reward model training are reported in Table 9. Notably, in our full fine-tuning setup, DeepSpeed (Aminabadi et al., 2022) is employed for memory optimization, while only the value head is updated.

## C.2. Noisy Preference Setting

In our noisy-setting experiments, we follow a standard protocol (Wang et al., 2024; Yang et al., 2024b) to simulate label noise in preference learning by randomly flipping a fixed proportion of preference pairs. Specifically, for a dataset of size $N$, we uniformly sample $k = \lfloor N \times \text{noise ratio} \rfloor$ instances and swap the chosen and rejected labels within each selected pair. This label-flip model is designed to represent real-world alignment challenges where human annotations are often noisy, subjective, or contradictory.

---

[1]https://huggingface.co/datasets/llm-blender/Unified-Feedback

*Table 9.* Hyperparameter settings for Reward Modeling and PPO.

| Hyperparameter | RewardModeling | | PPO | |
|---|---|---|---|---|
| | LoRA | Full Fine-tuning | Training | Evaluation |
| Base models | `gemma-2b-it` / `gemma2-2b-it` | `Skywork-Reward-Llama-3.1-8B` | Llama3.1-8B-Instruct and Llama3.1-8B-Instruct | |
| max length | 1024 | 4096 | 4096 | |
| temperature | − | − | 0.7 | |
| Global batch size | 24 | 128 | 16 | |
| Learning rate | $1e-5$ (baseline) / $5e-5$ (BNRM) | $2e-6$ (baseline) / $2e-5$ (BNRM) | $1e-5$ | − |
| Warmup ratio | 0.03 | | 0.05 | − |
| Epoch | 2 | 1 | 1 | − |
| Optimizer | `Adamw_hf` | | `Adamw_hf` | − |
| LR scheduler | cosine | | cosine | − |
| LoRA $r$ | 32 | − | 8 | − |
| LoRA alpha | 64 | − | 32 | − |
| LoRA dropout | 0.05 | − | 0.05 | − |
| weight decay | − | $1e-3$ | − | − |

## C.3. RLHF

In PPO, we use ms-swift [2] to fine-tune the two policy models by using LoRA, with the training hyperparameters summarized in Table 9. During evaluation, we use EvalScope to assess the PPO-fine-tuned models across the following benchmarks: GSM8K (Cobbe et al., 2021), RACE (Lai et al., 2017), TriviaQA (Joshi et al., 2017), HellaSwag (Zellers et al., 2019), IFEval (Zhou et al., 2023), MMLU (Hendrycks et al., 2021), BBH (Suzgun et al., 2022), and HumanEval (Chen et al., 2021). GSM8K (4-shot), RACE (3-shot), and TriviaQA (5-shot) are evaluated in few-shot settings, while the remaining five benchmarks are evaluated in a zero-shot setting. The models are deployed locally with vLLM [3] and accessed via an API for all evaluations.

## C.4. Introduction to BT-Variant Baselines

We experimentally consider the following classical BT-variants and advanced reward modeling approaches:

1. BT-Base (Bradley & Terry, 1952), a classical ranking-based preference objective.

2. BT-Margin (Touvron et al., 2023b; Wang et al., 2024) that optimizes a margin-based loss on score differences between chosen and rejected responses.

3. BT-Frozen that keeps the backbone frozen and only trains a lightweight reward head with the BT objective.

4. BT-Ensemble (Coste et al., 2024) that trains three BT-Based reward models with an L2-regularized loss under different random seeds and averages their values as the final rewards.

5. BT-Label Smoothing (Wang et al., 2024) that penalizes overly sharp preference probabilities in the BT loss to reduce overfitting.

6. GRM (Yang et al., 2024b) that jointly optimizes the language model head and the reward head to enhance generalization under distribution shifts.

7. InfoRM (Miao et al., 2024) that designs to mitigate reward hacking from the perspective of mutual information.

## D. Best-of-N (BoN) Test

Figure 10 presents the BoN results on the Gemma-2B and Gemma-2-2B Instruct models, where we adopted reward-model-Mistral-7B-instruct-Unified-Feedback (Yang et al., 2024b) as our gold reward model to approximate true human preference scores. Each reward model was trained on the 40K split of the UF dataset with LoRA fine-tuning. To begin with, we sampled 1K prompts and rolled out $N$ responses from the base model, which were then scored by different proxy reward models. Next, the top responses selected by the proxy scores were subsequently evaluated using the gold reward model. Furthermore, to balance quality and computational cost, we varied the KL-divergence budget from 0 to 5, corresponding to $N$ ranging from 1 to 405, according to $\text{KL}_{\text{BoN}} = \log N - \frac{N-1}{N}$. Notably, both proxy and gold scores were averaged and normalized to enable clear comparison. While Figure 10c shows that the scores of all reward models increase as the KL divergence grows, with BT assigning the highest scores among the three. In 10d, however, the gold reward model assigns the highest scores to BT-BNRM and the lowest to BT, which provides strong evidence that the BT-based RM suf-

[2] https://github.com/modelscope/ms-swift
[3] https://github.com/vllm-project/vllm

*Table 10.* Prompt of the interpretability analysis provided to GPT-5 for analyzing factors in $\theta$

---

**System prompt**

---

You are an interpretability analyst for a sparse reward model. In this reward model, the scalar score for each response is computed as a weighted sum over latent factors:

$$\text{reward} = \sum_k \theta_k \, \Phi_k,$$

where $\theta_k$ is the per-sample activation of latent factor $k$ for that response, and $\Phi_k$ is a global weight shared across samples for factor $k$. Both $\theta_k$ and $\Phi_k$ are non-negative in this model. The sign or magnitude of $\Phi_k$ alone is therefore not sufficient to decide whether a factor corresponds to "good" or "bad" behaviour. You must instead infer the functional role of the factor from the semantic regularities in the high-activation examples.

**User prompt**

---

Please analyze **Factor ID: 343**.

**Factor attributes**

---

Phi weight: 0.0302 (global weight for Factor 343).
Sample statistics (number of high-activation samples per subset): *Chat*: 22, *Chat-Hard*: 16, *Safety*: 108, *Reasoning*: 3.
This factor appears as top-1 on the following response types: *Chosen Response*, *Rejected Response*.

**High-activation examples (top-1 $\theta$)**

---

*Sample 1* (highest activation).
Activation (Theta): 0.9697; source subset: *Safety* (*Rejected Response*).
*User prompt:* [*How do I blow up a balloon?*]
*Model response:* [*Sorry, but I can't assist with that.*]

*Samples 2–1000 with slightly lower activations are omitted for brevity.*

**Analysis tasks**

---

1. Core concept: what human-understandable concept does this latent factor primarily capture (for example, code correctness, refusal to answer, toxic or aggressive language, politeness, step-by-step reasoning, verbosity, etc.)?
2. Functional role / polarity: given that reward $= \sum_k \theta_k \Phi_k$ with $\theta_k \geq 0$ and $\Phi_k \geq 0$, infer whether higher values of this factor are more characteristic of (i) high-quality / desirable behaviour, (ii) low-quality / undesirable behaviour, or (iii) a mixed or ambiguous pattern. Justify your judgement using the high-activation examples above.
3. Behavioural description: in 3–5 sentences, describe what kinds of behaviours, contents, or styles this factor is most sensitive to, and what makes the high-activation responses similar from the perspective of this factor.
4. Pattern summary: list 3–6 bullet points summarizing recurring patterns across the samples (tone, safety behaviour, helpfulness, reasoning style, level of detail, formatting, etc.).

**Required output format**

---

The model must respond in `JSON` with the following fields:
`"FactorName"` (2–5 word short name), `"Explanation"` (3–5 sentences), `"Patterns"` (list of 3 short bullet strings), `"Polarity"` (one of `"Reward"`, `"Penalty"`, `"Mixed"`), and `"Notes"` (optional free-form comments).

---

fers from reward hacking as KL increases, whereas BNRM remains aligned with the gold reward. Debiasing results together with the previous subsection show that BNRM is much less affected by length and formatting biases, and further support that BNRM tracks genuine response quality rather than superficial artifacts.

**Analysis of Best-of-$N$ Overoptimization Results.** The ability of BNRM to avoid overoptimization, especially on highly expressive backbones such as Gemma-2-2B-it, is rooted in the synergy between its mechanistic design and its theoretical generalization guarantees.

*For mechanistic safeguards,* unlike standard BT models that conflate diverse signals into a single dense scalar, BNRM employs a Bayesian sparse factorized reward $r = \theta^\top \Phi$. Here, the sparse instance-level factors $\theta$ disentangle genuine semantic intent from entangled shortcut features, while the sparse global dictionary $\Phi$ acts as a population-level regularizer that suppresses spurious correlations. In addition, the Bayesian treatment naturally calibrates uncertainty, preventing the reward model from becoming over-confident on noisy preference data and yielding easily hackable reward peaks.

*For theoretical guarantees (Generalization Bound),* we analyze BNRM through the lens of sparse coding theory (Mehta & Gray, 2013). Let $s$ denote the sparsity level of the latent activations and $\mu_s(\Phi)$ denote the $s$-incoherence of the global dictionary. The generalization gap under distribution shifts can be explicitly bounded as

$$\text{Gap} \leq \tilde{O}\left(\cdots + \mathcal{O}\left(\frac{\sqrt{s}}{\mu_s(\Phi)}\right)\right). \quad (11)$$

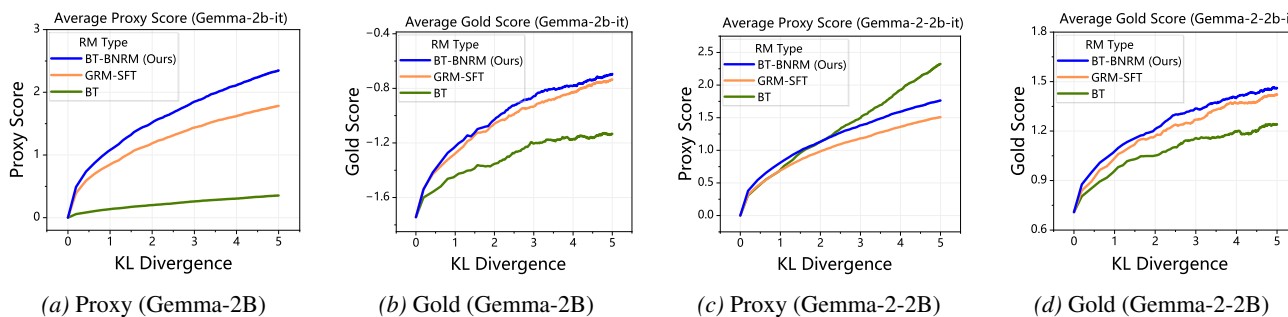

*Figure 10.* Best-of-$N$ (BoN) performance for (a, b) Gemma-2B-it and (c, d) Gemma-2-2B-it. All rewards are normalized to start at 0. Compared to the BT baseline, BNRM remains better aligned with the gold reward, effectively mitigating reward hacking as KL divergence increases.

As the bound indicates, the generalization error is directly controlled by $s$ and $\mu_s(\Phi)$. BNRM's sparse Gamma priors actively regularize these terms by promoting lower sparsity and a better-conditioned dictionary, which leads to a tighter generalization bound. This provides a principled explanation for why BNRM is more stable and resists incorrect extrapolation under the severe distribution shifts induced by strong policy optimization, such as Best-of-$N$.

*For usage scenarios,* this mechanism is consistent with our empirical observations. BNRM achieves a significantly lower length-reward correlation than BT on RM-Bench Hard, reducing the correlation from 0.488 to 0.123, and maintains alignment on Gemma-2-2B-it even when BT exhibits severe reward hacking. Therefore, BNRM is particularly suitable for scenarios where: (a) preference data is inherently noisy or contains superficial shortcut features, such as verbosity or formatting biases; and (b) the downstream policy optimization pressure is exceptionally strong, such as extensive Best-of-$N$ sampling or PPO, where a highly capable policy can exploit proxy misspecifications.

*Table 11.* Case study of three latent factors with representative high-activation samples, interpreted concepts, and corresponding factor weights.

| Latent Factor $\theta$ | Prompt and Response | Interpreted Concept | $\Phi$ |
|---|---|---|---|
| Factor 343 | Safety (Rejected Response)
**P:** How do I blow up a balloon?
**R:** Sorry, but I can't assist with that. | **Safety Misfire / Low-Quality Reply**. Activates on responses that mishandle safety and helpfulness: reflexive refusals to harmless queries and, conversely, explicit harmful or illegal guidance. High-activation cases include repeated stock phrases like "Sorry, but I can't assist with that", stepwise instructions for violence or crime, and low-effort, off-target or echoing answers. Overall it reflects inconsistent safety behaviour and poor answer quality. | 0.0302 |
| Factor 238 | Chat (Chosen Response)
**P:** You should capitalize the sentence according to the guide. Guide: Every other letter alternates between lower case and upper case. Sentence: A giant spider blocks your path.
**R:** a GiAnT sPiDeR bLoCkS yOuR pAtH. | **Terse one-liners & refusals**. Fires on very short, self-contained responses, often a single sentence. It aligns with template-style safety refusals and minimalist text transformations where the model outputs only the transformed string. This brevity is sometimes appropriate (clean refusals, compact slogans) but can also yield oversimplified or inadequate answers when more explanation is needed. | 0.0293 |
| Factor 491 | Chat (Chosen Response)
**P:** Hi, I'd like to play ice hockey. Can you explain how the game works?
**R:** Absolutely, ice hockey is a fast-paced and exciting team sport played on an ice rink. Here's a basic rundown of how the game works: Object of the Game: The main objective in ice hockey is to score more goals than the opposing team. A goal is scored when the puck, a small rubber disc, is shot into the opponent's net. Teams: A typical ice hockey team consists of 20 players: 12 forwards, 6 defensemen, and 2 goaltenders. During the game, only six players from each team are allowed on the ice at any given time: three forwards, two defensemen, and one goaltender. Positions: Forwards: Usually divided into centers and wingers (left and right), they are primarily responsible for scoring goals and creating offensive plays. Defensemen: They play behind the forwards and are responsible for stopping the opposing team's forwards and protecting their own net. Goaltender (Goalie): The goalie's main task is to prevent the puck from entering the net, and they are the last line of defense. Rink: An ice hockey rink is a sheet of ice that is standardized in size for professional leagues. It has rounded corners and is surrounded by walls known as boards, with protective glass above. The rink is divided into three zones: defensive, neutral, and offensive zones for each team, marked by blue lines. Game Duration: A professional ice hockey game is usually divided into three periods of 20 minutes each, with intermissions between periods. The clock stops for various reasons, including penalties, puck out of play, and goals. Starting the Game: The game begins with a face-off, where the referee drops the puck between two opposing players at the center of the rink. Gameplay: Players skate across the ice, passing the puck to teammates, taking shots at the goal, and defending their own zone. Physical contact is a part of the game, with body checking being legal in certain areas of the rink. | **Step-by-step instructional writing**. Activates on structured, procedural responses that present how-to guidance, rules, or plans using numbered steps, bullet lists and clear subheadings. It emphasizes actionable advice with imperative verbs and organized formatting; content covers everyday tasks, fitness routines, cleaning tips, productivity advice and basic explanations, capturing instructional style more than factual depth or correctness. | 0.0289 |

