# OpenReview forum: "Mitigating Reward Hacking in RLHF via Bayesian Non-negative Reward Modeling"
_ICML.cc/2026/Conference — ICML 2026 spotlight_

### Official Review · Reviewer_hKV5 · 2026-02-23

**Soundness:** 3
**Presentation:** 3
**Significance:** 3
**Originality:** 3
**Overall Recommendation:** 5
**Confidence:** 3

**Summary:**

BNRM proposes a Bayesian reward modeling framework that augments the BT model with a sparse, non-negative latent factorization to disentangle reward components while implicitly suppressing spurious biases. It uses instance-specific latent variables plus sparsity over global factors, with amortized variational inference for scalable, uncertainty-aware training, and is designed to be composable with existing reward model architectures (e.g., Llama-skyreward). The authors show comprehensive experiments showing reduced reward hacking (over-optimization) and improved robustness under distribution shift.

**Compliance With Llm Reviewing Policy:**

Affirmed.

**Final Justification:**

My concern has been addressed.

**Key Questions For Authors:**

- Figure 5 shows that compared to other methods, the proposed method yields relatively high scores in the shorter range (0-200). But in the hard arena, should those short passages actually score high?
- L209 lobal-> local ?
- Multiple citation
   - Coste, T., Anwar, U., Kirk, R., and Krueger, D. Reward model ensembles help mitigate overoptimization. arXiv preprint arXiv:2310.02743, 2023.
   - Coste, T., Anwar, U., Kirk, R., and Krueger, D. Reward model ensembles help mitigate overoptimization, 2024. URL https://arxiv.org/abs/2310.02743.
- What is the definition of d_model?
- Equation L 139 is hard to read; I think r_\phi alone would suffice.
- In L130, "r_\phi typically consists of a backbone feature extractor ....", It would be easier to read if you cite actual papers where "typically" is used instead of stating it here.

**Limitations:**

yes

**Strengths And Weaknesses:**

**Strengths**
- Offers a new Bayesian interpretation of BT model, making the reward modeling objective more principled and uncertainty-aware.
- The paper evaluates the proposed method across a wide range of experimental settings, and the results consistently support its effectiveness and robustness.
- BNRM can be integrated with existing reward model architectures; it appears broadly applicable and has strong potential for general use across different reward modeling setups.


**Weaknesses**
- Their approach uses Skywork-Reward-Llama-3.1-8B and SkyworkPreference-v0.2 to produce the results in Table 2. However, to my knowledge, Skywork-Reward-Llama-3.1-8B was trained using v0.1.
To ensure fairness, skyreward must also be compared to the version trained on dataset v0.2; otherwise, it is not fair.
- Section 5.3 shows improved benchmark accuracy and suggests enhanced robustness in BoN analysis (Appendix D), but it does not explain why the proposed method avoids overoptimization in these scenarios. (Figure 10: gemma-2b-it shows no overoptimization, but gemma-2-2b-it exhibits reward hacking in BT). Explaining this would help clarify when the proposed method should be used.

---

> ### Author Rebuttal · Authors · 2026-03-31
>
> **Response to Weakness 1 about the fairness of the Skywork comparison.** We appreciate the reviewer’s correction regarding the data versions. To ensure a fair comparison, we have re-evaluated ours against the official Skywork-Reward-Llama-3.1-8B-v0.2 baseline. Under this fair setting, the v0.2 baseline achieves 93.1, while BNRM reaches 93.6. This 0.5 gain on an already high-performing model shows that BNRM’s Bayesian sparse factorization provides genuine architectural improvements beyond data quality. We will update Tab.2 in the final version.
>
> **Response to Weakness 2 about Overoptimization and When to Use BNRM.** The ability of BNRM to avoid overoptimization—especially on highly expressive backbones like Gemma-2-2B-it—is rooted in the synergy between its mechanistic design and its theoretical generalization guarantees.
>
> *1. Mechanistic Safeguards:* Unlike standard BT models that conflate diverse signals into a single dense scalar, BNRM employs a Bayesian sparse factorized reward $r = \theta^\top \Phi$. Here, the sparse instance-level factors $(\theta)$ disentangle genuine semantic intent from entangled shortcut features, while the sparse global dictionary $(\Phi)$ acts as a population-level regularizer that suppresses spurious correlations. In addition, the Bayesian treatment naturally calibrates uncertainty, preventing the reward model from becoming overconfident on noisy preference data and yielding easily "hackable" reward peaks.
>
> *2. Theoretical Guarantee:* To formally ground this empirical robustness, we analyze BNRM through the lens of sparse coding theory [1]. Let $s$ denote the sparsity level of the latent activations and $\mu_s(\Phi)$ denote the $s$-incoherence of the global dictionary. The generalization gap under distribution shifts can be explicitly bounded as follows:
> $\text{Gap} \leq \tilde{O}\left(\cdots + \mathcal{O}\left(\frac{\sqrt{s}}{\mu_s(\Phi)}\right)\right)$.
> As the bound indicates, the generalization error is directly controlled by $s$ and $\mu_s(\Phi)$. BNRM’s sparse Gamma priors actively optimize these terms—promoting lower sparsity and a better-conditioned dictionary—which leads to a tighter generalization bound. This provides a principled explanation for why BNRM is more stable and resists extrapolating incorrectly under the severe distribution shifts induced by strong policy optimization (e.g., Best-of-N).
>
> *3. When should BNRM be used?* BNRM achieves a significantly lower length-reward correlation (0.123) compared to BT (0.488) on RM-Bench Hard, and maintains alignment on Gemma-2-2B-it even when BT exhibits severe reward hacking. Therefore, we recommend using BNRM in scenarios where: (a) Preference data is inherently noisy or contains superficial shortcut features (e.g., verbosity or formatting biases). (b) The downstream policy optimization pressure is exceptionally strong (e.g., extensive BoN sampling or PPO), where the policy is highly capable of exploiting proxy misspecifications. We will incorporate the analyses in the camera ready version.
>
> [1] Nishant Mehta, Alexander Gray. Sparsity-based generalization bounds for predictive sparse coding. ICML, 2013.
>
> **Response to Q1 about Arena Hard and token length.** We would like to clarify two key aspects:
>
> *(1) Experimental Context of Figure 5:* The results in Figure 5 were obtained from RM-Bench (Hard), not Arena Hard. This specific split is deliberately curated with rejected responses that are longer and better-formatted than the chosen ones to test for length/style bias. In this controlled setting, higher scores for shorter responses are precisely the intended and correct behavior, demonstrating that BNRM evaluates content quality rather than superficial length. BNRM achieves a significantly lower length-reward correlation (0.123) compared to the BT baseline (0.488), confirming its robustness.
>
> *(2) Performance on Arena-Hard:* To address the reviewer’s concern, we further evaluated our BNRM-aligned Llama-3.1-8B-Instruct on the Arena-Hard-v0.1. Under both GPT-4.1 and PhD-level expert judgment, ours achieves consistent superiority with a 50% win rate, 28% tie rate (GPT-4.1), 51% win rate and 18% tie rate (Human), significantly outperforming the base Llama-3.1-8B-Instruct. Notably, statistical analysis reveals that our model's average response length is 667.4 tokens, compared to the baseline's 852.8 tokens. These results indicate that BNRM prioritizes information density and conciseness, delivering high-quality, human-preferred answers without the redundant fillers often used to hack reward models. Detailed statistics and win-rate tables are available in Table 1 and Table 2 at https://anonymous.4open.science/r/BNRM-1353.
>
> **Response to Q2-Q6 about writing.** ''lobal'' in Line 209 should be ''Global''. We clarify that $d_{model}$ refers to the hidden state dimensionality of the backbone LLM (e.g., 2048 for Gemma-2B and 4096 for Llama-3.1-8B). We will fix issues in the camera-ready version.

---

> > ### Author Rebuttal · Reviewer_hKV5 · 2026-04-01
> >
> > Thank you for the rebuttal. My concern has been addressed. I'll keep score.

---

> > > ### Author Response · Authors · 2026-04-01
> > >
> > > Dear reviewer, we are glad to hear that we successfully addressed all your concerns. Thanks for your constructive suggestions, and we will incorporate them in our camera-ready.

---

### Official Review · Reviewer_24Ao · 2026-02-24

**Soundness:** 4
**Presentation:** 4
**Significance:** 3
**Originality:** 3
**Overall Recommendation:** 5
**Confidence:** 3

**Summary:**

This paper proposes the Bayesian Non-negative Reward Model (BNRM), a framework that integrates Non-negative Factor Analysis (NFA) into the Bradley-Terry preference model using amortized variational inference. By decomposing rewards into sparse, non-negative latent factors, the method aims to improve interpretability and mitigate reward hacking driven by spurious correlations (e.g., length bias) in RLHF.

**Compliance With Llm Reviewing Policy:**

Affirmed.

**Final Justification:**

my concerns have been addressed

**Key Questions For Authors:**

1.The assumption that rewards can be decomposed into a linear dot product of non-negative factors may limit the model's ability to capture complex, non-monotonic preference interactions compared to dense, non-linear heads.

2.Ablation studies reveal that the model’s performance is highly sensitive to the KL divergence coefficient ($\eta$), which may complicate training stability and reproducibility across different datasets.

3.The use of variational inference with custom Weibull reparameterization introduces additional implementation complexity and potential training instability compared to standard cross-entropy or margin losses.

**Limitations:**

The authors have not adequately discussed the limitations of their work, and the impact statement is generic. I suggest the following improvements:

1.  **Explicit Limitations Section:** The paper would benefit from a dedicated discussion on the limitations of the BNRM framework. Specifically, the authors should address:
    *   **Expressivity:** Whether the structural constraint of a non-negative linear combination ($r = \theta^T \Phi$) limits the model's ability to approximate highly complex or non-monotonic reward landscapes compared to dense, deep neural network heads.
    *   **Training Stability & Hyperparameters:** While Appendix A.3 shows sensitivity to the KL coefficient $\eta$, this should be acknowledged as a limitation in the main text. Variational inference is often harder to tune than standard objectives; discussing the stability of the Weibull parameterization would be valuable.

2.  **Societal Impact:** The current Impact Statement is boilerplate ("none of which we feel must be specifically highlighted here"). Since this work focuses on RLHF and alignment, the authors could briefly acknowledge the dual-use potential: while BNRM improves safety and alignment, robust reward optimization techniques can theoretically be applied to optimize models for harmful or malicious objectives more effectively.

**Strengths And Weaknesses:**

**Strengths**

1.The integration of NFA with deep learning via a Weibull-parameterized variational inference network offers a theoretically sound approach to modeling uncertainty and sparsity in reward learning.

2.The method demonstrates superior performance over strong baselines (including ensembles and GRM) on OOD benchmarks like RewardBench, showing significant resilience to label noise and distribution shifts.

3.The sparsity constraints effectively act as a regularizer against spurious correlations, as evidenced by the substantial reduction in length bias on RM-Bench and the semantic interpretability of the learned factors.

---

> ### Author Rebuttal · Authors · 2026-03-31
>
> **Response to Q1 about method capability.** To clarify our design philosophy, BNRM operates through a synergy between a highly non-linear LLM backbone and a structured Bayesian output layer. While the final reward aggregation $r = \theta^\top \Phi$ is a linear dot product, the mapping from raw input $(x, y)$ to the local latent variables $\theta$ is performed by the deep backbone, which is fully capable of capturing intricate and non-monotonic preference interactions. By enforcing non-negativity and sparsity only at this final layer, we ensure that the captured signals remain human-interpretable without sacrificing the underlying feature extraction power of the model. In fact, ours maintains structural parity with BT models, which typically project dense hidden representations $z$ to a scalar reward via a single linear head $W_{bt}$. By reconfiguring this projection into a factorized form, BNRM does not fundamentally restrict expressiveness compared to mainstream reward models; rather, it introduces a principled inductive bias that enhances robustness. This architectural choice is further supported by the Linear Representation Hypothesis [1], which suggests that high-level semantic concepts are often approximately linearly encoded within LLM latent spaces. Consequently, a structured linear head at the output stage is often sufficient to recover these concepts while providing the added benefit of disentanglement. Our empirical findings strongly validate that this factorized formulation preserves sufficient expressivity while offering superior generalization. As shown in Tab. 1 and Fig. 4, BNRM consistently outperforms the dense BT baseline, including complex reasoning and chat-hard tasks. The performance gap remains substantial even under 40% label noise, indicating that the non-negative factorization acts as a robust regularizer that filters out spurious shortcuts without compromising the model’s ability to capture genuine human intent.
>
> [1] Park, Kiho, Yo Joong Choe, and Victor Veitch. The linear representation hypothesis and the geometry of large language models. ICML, 2024.
>
> **Response to Q2 about KL coefficient.** While performance varies with KL coefficient $\eta$ in Fig. 7 of Appendix, BNRM remains reasonably robust across a wide range of values. Crucially, we used a fixed setting of $\eta=10^{-5}$ across all datasets without per-dataset tuning. To further address your concern, we additionally compare our results at $\eta = 10^{-6}$ against BT, which corresponds to the weakest-performing setting in our experiments. As shown in Table 4 at https://anonymous.4open.science/r/BNRM-1353, even at this setting, BNRM's RewardBench accuracy (72.7%) remains substantially higher than BT's (64.5\%).
>
> **Response to Q3 about implementation complexity and training stability.**
> *(1) Implementation Complexity:* We agree that BNRM introduces additional structure compared to the deterministic BT, but this overhead is remarkably lightweight. As discussed in Appendix A.2, we provided a detailed complexity analysis, where the additional computational overhead introduced by BNRM is limited relative to backbone. To further validate this efficiency, we report the wall-clock training time for full-parameter fine-tuning on the Skywork-Reward-Preference-v0.2 dataset in Tab. 6 at https://anonymous.4open.science/r/BNRM-1353. For the Gemma-2B-it model, BNRM increases training time by only 7.7% (from 3.88h to 4.18h). For the larger Llama-3.1-8B-Instruct model, the overhead is even more minimal, representing a mere 1.3% increase (from 11.70h to 11.86h). This trend indicates that as the model scale increases, the marginal cost of BNRM becomes nearly imperceptible, making it highly suitable for modern large-scale RLHF workflows.
>
> *(2) Training Stability:* The Weibull distribution is specifically chosen for its ability to model sparse, positive latent variables while admitting an efficient and stable reparameterization path. Despite the use of variational inference and Weibull reparameterization, BNRM still exhibits training stability. Our localized uncertainty modeling—confining stochasticity to the output factors—prevents training instability from propagating through the backbone. As shown in Fig. 8, BNRM achieves a validation accuracy of 71.75% within just 0.25 epochs, a level that standard BT and GRM variants struggle to reach even after 1.5 epochs. Throughout our experiments, BNRM converged to a higher and more stable validation plateau. This suggests that rather than introducing difficulty, the non-negative sparsity constraints act as an effective regularizer, filtering out biased preference signals and leading to a more reliable optimization trajectory.
>
> **Response to Limitations**. We agree that the limitations and broader impact of our work should be discussed more explicitly, and we will include these insightful and deep discussions to improve both parts in our camera-ready.

---

> > ### Author Rebuttal · Reviewer_24Ao · 2026-04-02
> >
> > I would like to thank the authors for their rebuttal. Now my concerns have been addressed.

---

> > > ### Author Response · Authors · 2026-04-05
> > >
> > > Dear reviewer, we are glad to hear that we successfully addressed all your concerns. Thanks for your constructive suggestions, and we will incorporate them in our camera-ready.

---

### Official Review · Reviewer_xtVx · 2026-03-09

**Soundness:** 3
**Presentation:** 4
**Significance:** 3
**Originality:** 4
**Overall Recommendation:** 5
**Confidence:** 4

**Summary:**

This paper proposes Bayesian Non-negative Reward Modeling (BNRM), a reward modeling method designed to address the issue of spurious preferences in Bradley–Terry (BT) preference models. Through a sparse and non-negative latent factor generative process combined with variational inference, BNRM significantly improves reward modeling performance. Experimental results in RLHF further demonstrate that BNRM not only enhances preference modeling but also improves LLM alignment by mitigating reward hacking and disentangling latent reward factors.

**Compliance With Llm Reviewing Policy:**

Affirmed.

**Key Questions For Authors:**

Overall, I find the paper well-written and the proposed method interesting. I only have a few clarification questions:

1. In the noisy setting experiments, how is the noise introduced into the preference data? More details on the noise generation process would help clarify the experimental setup.

2. How does BNRM compare with other uncertainty-aware reward modeling approaches? It would be helpful to discuss the differences and potential advantages over existing uncertainty-based reward models.

3. The paper claims that the rewards learned by BNRM are interpretable. Could the authors elaborate on why the learned rewards are considered interpretable, and whether this interpretability is demonstrated or validated in the experiments?

**Limitations:**

yes

**Strengths And Weaknesses:**

Strengths:
1. Rewards are decomposed into several latent reward factors, which help eliminate the effect of spurious signal introduced by annotators or irrelevant sources.

2. Traditional BT RMs mix different reward factors but the sparse reward factors in BNRM help improve interpretability of the final rewards.

3. The proposed method is not only effective in preference modeling but also applicable in downstream LLM RLHF fine-tuning.

Weaknesses:
1. Although the paper claims improved interpretability, the learned latent reward factors may not be stable. Different training runs could potentially produce different factor decompositions, raising concerns about the robustness of the interpretability claim.

---

> ### Author Rebuttal · Authors · 2026-03-31
>
> **Response to W1 and Q3 about interpretability.** We thank the reviewer for the opportunity to elaborate on the interpretability of BNRM. Our model's interpretability is not a post-hoc artifact but is mathematically rooted in its architecture and rigorously validated through multiple empirical lenses.
>
> *(1) Theoretical Guarantee (Identifiability and Transparent Mechanics):* Unlike standard dense RMs, BNRM enforces strict non-negativity and sparsity ($r = \theta^\top \Phi$). By formulating this as a non-negative matrix factorization (NMF) task, we leverage the theory of partial identifiability [1] to guarantee that our learned factors correspond to unique, recoverable semantic structures rather than arbitrary mathematical noise. Furthermore, non-negativity removes "sign ambiguity" (where negative weights in linear models conflate penalties for good features vs. rewards for bad ones), making the functional role of each factor mathematically transparent.
>
> *(2) Empirical Validation I (Semantic Stability):* To prove these factors represent stable, human-understandable concepts, our stability analysis ((Table 3 in https://anonymous.4open.science/r/BNRM-1353) demonstrates that 7 core "Semantic Families" (e.g., Safety Refusal, Step-by-step Guidance) are robustly recovered across varying seeds, model scales, and datasets. Moreover, Table 8 in Appendix directly links representative factors to specific behavioral concepts via their high-activation samples.
>
> *(3) Empirical Validation II (Mechanistic Error Rectification):* This interpretability allows us to mechanically trace how the model avoids reward hacking. As shown in Fig. 6, we observe a clear "Error Rectification" mechanism. When instance-level activations ($\theta$) mistakenly favor a rejected response, the global sparsity of $\Phi$ effectively suppresses these signals by driving their weights toward zero—a mechanistic intervention we observed in 25.5% of test samples.
>
> *(4) Empirical Validation III (Unsupervised Debiasing):* The "parts-based" representation naturally ignores superficial shortcuts. This is visually evident in Figs. 5 and 9, where BNRM evaluates content quality rather than verbosity, achieving a significantly lower length-reward correlation (r=0.123) compared to the standard BT model (r=0.488).
>
> [1] Gillis, Nicolas, and Róbert Rajkó. "Partial identifiability for nonnegative matrix factorization." SIAM Journal on Matrix Analysis and Applications 44, no. 1 (2023): 27-52.
>
> **Response to Q1 about noisy setting.** In our noisy-setting experiments, we follow a standard protocol [1,2] to simulate label noise in preference learning by randomly flipping a fixed proportion of preference pairs. Specifically, for a dataset of size $N$, we uniformly sample $k = \lfloor N \times \text{noise ratio} \rfloor$ instances and swap the 'chosen' and 'rejected' labels within each selected pair. This label-flip model is designed to represent real-world alignment challenges where human annotations are often noisy, subjective, or contradictory. We will include a detailed description in the camera-ready version.
>
> [1] Wang B, Zheng R, Chen L, et al. Secrets of rlhf in large language models part ii: Reward modeling[J]. arXiv preprint arXiv:2401.06080, 2024.
>
> [2] Yang, R., Ding, R., Lin, Y., Zhang, H., and Zhang, T. Regularizing hidden states enables learning generalizable reward model for llms. NeurIPS, 2024.
>
> **Response to Q2 about uncertainty-aware baselines.**  We thank the reviewer for this insightful question. Among our baselines, InfoRM is indeed the primary uncertainty-aware method. However, BNRM offers a fundamentally different and conceptually stronger approach to uncertainty modeling. While InfoRM mitigates overoptimization indirectly via a variational information bottleneck and latent outlier signals, BNRM explicitly models uncertainty within a structured reward decomposition $r(x,y)=\theta^\top\Phi$. By applying a Bayesian treatment to both instance-level factors ($\theta$) and the global dictionary ($\Phi$), BNRM goes beyond simply penalizing uncertainty; it provides a disentangled, interpretable representation that directly prevents the model from becoming overconfident on specific spurious features. This structural advantage translates to consistent empirical gains across multiple dimensions. In direct reward modeling, BNRM consistently outperforms InfoRM on both ID and OOD evaluations in Tab. 1, including UF, HHH, MT, and RewardBench. Furthermore, this superiority extends to downstream policy alignment, where Table 3 demonstrates stronger RLHF performance on both Llama-3.1-8B-Instruct and OpenRLHF-Llama3-8B-SFT. Finally, our formulation yields stronger unsupervised length debiasing, as confirmed by Fig. 5. Together, these results confirm that BNRM provides a more interpretable formulation and stronger practical robustness than existing uncertainty-aware baselines.

---

> > ### Author Rebuttal · Reviewer_xtVx · 2026-04-02
> >
> > I would like to thank the authors for their rebuttal. Now my concerns have been addressed.

---

> > > ### Author Response · Authors · 2026-04-02
> > >
> > > Dear reviewer, we are glad to hear that we successfully addressed all your concerns. Thanks for your constructive suggestions, and we will incorporate them in our camera-ready.

---

### Official Review · Reviewer_L5nr · 2026-03-13

**Soundness:** 3
**Presentation:** 3
**Significance:** 3
**Originality:** 3
**Overall Recommendation:** 5
**Confidence:** 2

**Summary:**

This paper proposes BNRM, a Bayesian non-negative reward model for RLHF, with the goal of mitigating reward hacking and improving generalization. Instead of assigning each response a single deterministic reward score, the method decomposes reward into sparse, non-negative latent factors, including instance specific local factors and dataset level global factors, so that the model is less likely to rely on superficial cues such as length or formatting. Technically, it is built on the Bradley-Terry model, uses Gamma priors and Weibull variational inference to learn these latent reward factors, and is trained with an ELBO objective. Experiments show that BNRM improves reward modeling performance, robustness to noisy or limited data, out-of-distribution generalization, and stability under PPO based optimization, while also offering some interpretability at the factor level.

**Compliance With Llm Reviewing Policy:**

Affirmed.

**Final Justification:**

My concerns have been addressed and I have raised my score.

**Key Questions For Authors:**

1. The paper’s central claim is that BNRM mitigates reward hacking, but the strongest evidence appears to rely on proxy-based evaluation rather than direct human assessment. Could the authors clarify whether they have any human evaluation, or any stronger evidence that the improved proxy behavior corresponds to genuinely better alignment with human preferences rather than better agreement with another reward model?
2. The method is motivated by the idea that the latent non-negative factors disentangle useful reward-relevant signals from spurious ones, but it is not fully clear how stable or semantically meaningful these learned factors are across runs, models, or datasets. Could the authors clarify whether the factorization is reproducible and whether similar factors emerge consistently?
3. Could the authors comment on whether BNRM is robust to hyperparameters, or whether it requires substantial tuning to work well?
4. The paper shows improvements in benchmark-based RLHF experiments, but it is still unclear how well the method would transfer to larger-scale or more open-ended RLHF settings, where reward hacking may arise in more complex ways than the current setup captures. Could the authors discuss the main barriers to scaling BNRM to such settings?

**Limitations:**

Yes.

**Strengths And Weaknesses:**

**Soundness.** The paper is technically solid overall. The main method is clearly motivated by the limitations of standard Bradley-Terry reward models, and the proposed Bayesian non-negative factorization is not just conceptual but implemented with a concrete variational-inference scheme using reparameterizable Weibull posteriors and KL regularization. The empirical section is also fairly broad, covering in-distribution and out-of-distribution reward modeling, low-resource and noisy settings, PPO-based RLHF, and debiasing analyses on length and formatting bias.

**Presentation.** The paper is generally well-organized and easy to follow. The motivation is clear, the contrast with standard scalar reward models is intuitive, and the figures help explain the local and global latent factors and their intended roles in disentanglement and debiasing. The appendix also includes useful implementation details and algorithmic steps.

**Significance.** Reward hacking in RLHF is an important problem, and the paper tackles it at the reward-model level rather than only through policy-side regularization. That makes the contribution potentially meaningful, especially because the method appears to be usable as a fairly general plug-in replacement for existing reward models and shows gains under distribution shift, label noise, and RLHF fine-tuning. However, the practical impact is somewhat constrained by the evaluation scope. Most of the evidence is still benchmark-based, and it remains uncertain how much the method would help in larger-scale production RLHF pipelines or under more open-ended human preference data where the failure modes may be broader than length and formatting bias.

**Originality.** The paper is meaningfully original in how it brings together non-negative factor analysis, Bayesian uncertainty modeling, and reward modeling for RLHF. The idea of replacing a dense scalar reward head with sparse non-negative latent factors is distinctive, and the interpretation of local sparsity as disentanglement and global sparsity as debiasing is a nice conceptual contribution. The interpretability analysis is also a useful addition.

---

> ### Author Rebuttal · Authors · 2026-03-31
>
> **Response to Q1 about Human Evaluation.** To bridge this gap, we provide a two-fold validation using both high-correlation benchmarks and direct human assessment.
>
> (1) We use Arena-Hard-v0.1 benchmark, specifically designed to yield the highest correlation with the Chatbot Arena [1]. Using GPT-4.1 as a judge, our BNRM-aligned Llama-3.1-8B-Instruct achieves a 50% win rate and 28% tie rate against the base Llama-3.1-8B-Instruct, significantly outperforming the baseline (22% win rate).
>
> (2) To provide direct evidence, we conducted a blind pairwise human evaluation. Two experts with PhD backgrounds evaluated 50 randomly sampled pairs from Arena-Hard. Our model achieved a 51% win rate and 18% tie rate, consistently mirroring the trends observed in our LLM-as-a-judge results. These consistent gains across both automated and human rubrics demonstrate that BNRM’s mitigation of reward hacking translates into genuine improvements in response quality, rather than merely overfitting to another reward model. More detailed results are reported in Table 1 at https://anonymous.4open.science/r/BNRM-1353.
>
> [1] Li T, Chiang W L, Frick E, et al. From crowdsourced data to high-quality benchmarks: Arena-hard and benchbuilder pipeline[J]. arXiv, 2024.
>
> **Response to Q2 about reproducibility of factorization.**
>
> *(1) Theory Analysis:* The reward in BNRM is constructed as
> $r(x, y) = \theta_v^{\top} \Phi,$
> where $\theta_v$ denotes the instance-specific latent factor and $\Phi$ denotes the global reward dictionary. It can be formulated as constrained non-negative matrix factorization (NMF) task: $\min_{\theta \ge 0,\Phi \ge 0} \lVert r-\theta^\top\Phi \rVert_F^2$. Following [1], such factorizations are partially identifiable under two strict conditions: (a) a Selective Window (i.e., there exists a row $j$ such that $\Phi(j,:) = \alpha e_{(k)}^\top$ for some $\alpha>0$, meaning a specific factor is uniquely activated by a distinct pattern), and (b) a Sparsity Constraint. BNRM naturally satisfies both conditions through its non-negative architecture and Weibull-parameterized sparse priors. This mathematically guarantees the decomposition is unique up to permutation and scaling, ensuring the factors correspond to distinct structural properties in the preference data.
>
> [1] Gillis, Nicolas, and Róbert Rajkó. "Partial identifiability for nonnegative matrix factorization." SIAM Journal on Matrix Analysis and Applications 44, no. 1 (2023): 27-52.
>
> *(2) Empirical Reproducibility (Semantic Stability):* We further conducted  stability tests across multiple random seeds, model scales, and datasets (see Tab.3 at https://anonymous.4open.science/r/BNRM-1353). While exact factor indices may permute, we robustly recover 7 core "Semantic Families" (e.g., Safety Refusal, Concise Problem-solving) across all settings. For instance, the "Safety Refusal" concept (Factor 343 in our main run) consistently emerges with the exact same behavioral triggers in other seeds (matching factors 205/823/917). This cross-run consistency is anchored by the global dictionary $\Phi$, which acts as a population-level regularizer to suppress noise and ensure reproducible disentangled representations.
>
> **Response to Q3 about hyperparameters.**  BNRM is robust and requires minimal tuning. The only method-specific hyperparameter is the KL coefficient $\eta$ (fixed at $\eta = 10^{-5}$); other settings follow GRM. Ablations (App. Fig. 7) show performance variation across $\eta$ remains moderate. Crucially, fixing $\eta=10^{-5}$ universally achieves strong performance across all datasets without per-dataset tuning.
>
> **Response to Q4 about larger-scale and open-ended RLHF settings.** We agree that scaling RLHF to open-ended settings introduces complex reward hacking challenges.
>
> *(1) Computational Scalability:* Scaling any RLHF pipeline inherently demands significant resources for the LLM backbone. However, the marginal barrier introduced by BNRM is negligible. As detailed in our response to Reviewer 24Ao Q3, our Bayesian layer adds only a 1.3% training time overhead on an 8B model. Thus, BNRM scales seamlessly alongside standard reward models without imposing new computational bottlenecks.
>
> *(2) Algorithmic Robustness:* The increased complexity of reward hacking at scale is precisely where BNRM provides the most value. Under strong optimization pressure, standard dense reward heads are highly vulnerable to proxy misspecifications because they conflate genuine intent with complex shortcuts. In contrast, BNRM’s sparse factorized formulation ($r=\theta^\top\Phi$) disentangles these signals and mathematically tightens the generalization bound under distribution shifts. This ensures resistance to overoptimization in complex, open-ended trajectories, as empirically validated by our OOD, noisy-label, and Arena-Hard results (Q1).

---

> > ### Author Rebuttal · Reviewer_L5nr · 2026-04-02
> >
> > Thanks the authors for their rebuttal. My concerns have been addressed and I have raised my score.

---

> > > ### Author Response · Authors · 2026-04-02
> > >
> > > Dear reviewer, we are glad to hear that we successfully addressed all your concerns and that you have raised your score. Thanks for your constructive suggestions, and we will incorporate them in our camera-ready.

---

### Decision · Program_Chairs · 2026-04-30

**Decision:**

Accept (spotlight)

**Comment:**

All reviewers agree on the high quality of the paper and that it should be accepted. Reviewers note that the paper is technically solid, with a clear Bayesian formulation and a coherent variational inference implementation. It shows consistent gains in reward modeling, robustness, and RLHF alignment across noisy, OOD, and debiasing settings. The rebuttal addressed the main concerns on human evaluation, interpretability, reproducibility, and scalability, leaving no major unresolved issues.